# Factors controlling marine aerosol size distributions and their climate effects over the Northwest Atlantic Ocean region

Betty Croft[1], Randall V. Martin[2,1], Richard H. Moore[3], Luke D. Ziemba[3], Ewan C. Crosbie[3,4],

Hongyu Liu[5], Lynn M. Russell[6], Georges Saliba[6], Armin Wisthaler[7,8], Markus Müller[7],

Arne Schiller[7], Martí Galí[9], Rachel Y.-W. Chang[1], Erin E. McDuffie[1,2], Kelsey R. Bilsback[10],

and Jeffrey R. Pierce[10]

[1]Department of Physics and Atmospheric Science, Dalhousie University, Halifax, NS, Canada

[2]McKelvey School of Engineering, Washington University in St. Louis, St. Louis, MO, USA

[3]NASA Langley Research Center, Hampton, VA, USA

[4]Science Systems and Applications, Inc., Hampton, VA, USA

[5]National Institute of Aerospace, Hampton, VA, USA

[6]Scripps Institute of Oceanography, University of California, San Diego, La Jolla, CA, USA

[7]Institute for Ion Physics and Applied Physics, University of Innsbruck, Technikerstrasse 25,

6020 Innsbruck, Austria

[8]Department of Chemistry, University of Oslo, P.O. 1033 – Blindern, 0315 Oslo, Norway

[9]Barcelona Supercomputing Center (BSC)

[10]Department of Atmospheric Science, Colorado State University, Fort Collins, CO, USA

*Correspondence to:* Betty Croft (betty.croft@dal.ca)

**Abstract.**

Aerosols over Earth's remote and spatially extensive ocean surfaces have important influences on planetary climate. However, these aerosols and their effects remain poorly understood, in part due to the remoteness and limited observations over these regions. In this study, we seek to understand factors that shape marine aerosol size distributions and composition in the Northwest Atlantic Ocean region. We use the GEOS-Chem-TOMAS model to interpret measurements collected from ship and aircraft during the four seasonal campaigns of the North Atlantic Aerosols and Marine Ecosystems Study (NAAMES) conducted between 2015 and 2018. Observations from the NAAMES campaigns show enhancements in the campaign-median number of aerosols with

diameters larger than 3 nm in the lower troposphere (below 6 km), most pronounced during the phytoplankton bloom maxima (May/June) below 2 km in the free troposphere. Our simulations, combined with NAAMES ship and aircraft measurements, suggest several key factors that contribute to aerosol number and size in the Northwest Atlantic lower troposphere, with significant regional-mean (40-60 $^{\circ}$N, 20-50 $^{\circ}$W) aerosol-cloud albedo indirect effects (AIE) and direct radiative effects (DRE) during the phytoplankton bloom. These key factors and their associated simulated radiative effects in the region include: (1) particle formation near and above the marine boundary layer (MBL) top (AIE: -3.37 W m$^{-2}$, DRE: -0.62 W m$^{-2}$), (2) particle growth due to marine secondary organic aerosol (MSOA) as the nascent particles subside into the MBL, enabling them to become cloud-condensation-nuclei-size particles (AIE: -2.27 W m$^{-2}$, DRE: -0.10 W m$^{-2}$), (3) particle formation/growth due to the products of dimethyl sulfide, above/within the MBL (-1.29 W m$^{-2}$, DRE: -0.06 W m$^{-2}$), (4) ship emissions (AIE: -0.62 W m$^{-2}$, DRE: -0.05 W m$^{-2}$) and (5) primary sea spray emissions (AIE: +0.04 W m$^{-2}$, DRE: -0.79 W m$^{-2}$). Our results suggest that a synergy of particle formation in the lower troposphere (particularly near and above the MBL top) and growth by MSOA contributes strongly to cloud-condensation-nuclei-sized particles with significant regional radiative effects in the Northwest Atlantic. To gain confidence in radiative effect magnitudes, future work is needed to understand 1) the sources and temperature-dependence of condensable marine vapors forming MSOA, 2) primary sea spray emissions, and 3) the species that can form new particles in the lower troposphere and grow these particles as they descend into the marine boundary layer.

## 1. Introduction

Marine atmospheric particles have important roles in Earth's climate system. Similar to particles in other regions, marine aerosols scatter and absorb solar radiation (Charlson et al., 1992), and modify cloud properties by acting as the seeds for cloud droplet formation (Boucher and Haywood, 2000; Lohmann and Feichter, 2005). Aerosols in the atmosphere's marine boundary layer (MBL) strongly influence the highly prevalent, low-altitude marine clouds, which have key climate cooling effects due to their reflection of incoming solar radiation (Wood, 2012; Chen et al., 2014). However, there remains high uncertainty about the magnitude of these aerosol effects (IPCC, 2013), due in part to limited understanding about the processes that control aerosols over Earth's expansive and remote ocean surfaces (Willis et al., 2018). Marine aerosols are strongly influenced

by natural, but poorly understood sources, making a large contribution to uncertainty in aerosol-
climate effects (Carslaw et al., 2010; Carslaw et al., 2013). Limited observations of aerosols and
their precursors over Earth's remote marine regions contribute to these knowledge gaps. In this
study, we focus on investigation of several factors controlling the seasonal cycle of aerosol size
and number and their resultant climate effects over the Northwest Atlantic Ocean.

Aerosol particles in the remote MBL have several seasonally varying sources (O'Dowd et al.,
2004; Leck and Bigg, 2005; de Leeuw et al., 2011; Karl et al., 2012). Primary particles are emitted
through wave breaking and bubble bursting processes that eject sea spray aerosols (SSA) of sea
salt and organic composition (Russell et al., 2010; de Leeuw et al., 2011; Ovadnevaite et al., 2011;
Gantt and Meskhidze, 2013; Prather et al., 2013; Hamacher-Barth et al., 2016; Brooks and
Thornton, 2018). SSA have a not-yet-well-understood dependence on wind speed (Monahan et al.,
1983; O'Dowd et al., 1997; Ovadnevaite et al., 2012; Grassian et al., 2015; Brooks and Thornton,
2018; Saliba et al., 2019) and sea surface temperature (Mårtensson et al., 2003; Jaeglé et al., 2011;
Kasparian et al., 2017; Saliba et al., 2019).  For the North Atlantic, observations indicate that
primary SSA make a limited (less than 30%) contribution to cloud condensation nuclei (CCN)
(Quinn et al., 2017, Zheng et al., 2018; Quinn et al., 2019) with no direct connection between SSA
emissions and plankton ecosystems because the organic SSA appears to arise from the ocean's
large pool of dissolved organic carbon (Quinn et al., 2014; Bates et al., 2020). SSA, however,
could modify the CCN number that activate to form cloud droplets (Fossum et al., 2020), act as
ice nuclei (Wilson et al., 2015; DeMott et al., 2016; Irish et al., 2017), and be more closely linked
with biogenic activity in other regions (Ault et al., 2013; Cravigan et al., 2015; O'Dowd et al.,
2015; Quinn et al., 2015; Wang et al., 2015; Schiffer et al., 2018; Christiansen et al., 2019;
Cravigan et al., 2019). Recent studies have highlighted knowledge gaps related to sea spray
emissions, particularly as related to the submicron sizes (e.g., Bian et al., 2019; Regayre et al,
2020). Measurement and modeling studies are needed to better understand and simulate the size-
resolved contribution of sea spray to the Northwest Atlantic MBL.

For the North Atlantic, secondary aerosol of biogenic origin is observed to be an important
seasonally varying contributor to marine particles and their growth to yield CCN (Sanchez et al.,
2018). Marine secondary aerosol can arise from the condensation of a variety of marine-vapor-
oxidation products, which form and grow particles (Ceburnis et al., 2008; Rinaldi et al., 2010;
Decesari et al., 2011). Formation of new aerosol particles in the marine environment is observed
to be favored in clean atmospheric layers just below the marine inversion and also above the MBL
top  (Kazil et al., 2011; Takegawa et al., 2020). Newly formed particles, including those from the
free troposphere can grow to CCN sizes (diameters larger than about 50 nm) through the
condensation of available organic and sulfur-containing vapors on descent into the MBL
(Korhonen et al., 2008). Once the particles reach CCN sizes, cloud processing (including aqueous
phase aerosol production, and cloud droplet coagulation with other droplets and interstitial
aerosols) also contributes to shaping the size distribution (Hoppel et al., 1986; Hoose et al., 2008;
Pierce et al., 2015). For the North Atlantic MBL, entrainment of growing new particles formed in
the relatively cleaner free troposphere is an important contributor to MBL particle number (Quinn
et al., 2017; Sanchez et al., 2018; Zheng et al., 2018). In the pristine conditions of the summertime
Arctic, both new particle formation (NPF) and growth (by condensation of organic and sulfur-
containing vapors) are frequently observed within the boundary layer itself (Leaitch et al., 2013;
Croft et al., 2016a; Willis et al., 2016; Collins et al., 2017; Burkart et al., 2017b). In addition to
sulfuric acid, other vapors including amines, methane sulfonic acid (MSA), ammonia, and iodine
all contribute to NPF in marine regions (O'Dowd, 2002; Facchini et al., 2008; Allan et al., 2015,
Chen et al., 2016; Croft et al., 2016a; Dall´Osto et al., 2018). Interpretation of a combination of
aircraft and ship-board observations with a size-resolved aerosol microphysics model is needed to
develop understanding of the relative importance of near and above MBL top NPF as a contributor
to aerosol size distributions in the Northwest Atlantic MBL.

Dimethyl sulfide (DMS) is one of the key contributors to secondary particle formation and growth
that is released from the oceans as a result of marine biogenic activity (Lana et al., 2012a; Galí and
Simó, 2015; Sanchez et al., 2018). The oxidation products of DMS include sulfuric acid and MSA
(Barnes et al., 2006), which can form new particles and grow existing particles to sizes that can
act as CCN (Hoffman et al., 2016; Hodshire et al., 2019). As well, hydroperoxymethyl thioformate
(HPMTF) is a recently discovered DMS-oxidation product, which could also contribute to NPF
and growth (Veres et al., 2020). The role of DMS in the climate system has undergone much debate
since 1987 when the CLAW hypothesis proposed that DMS could act as a regulator in a warming
climate (Charlson et al., 1987). For the North Atlantic and Arctic, observations have linked DMS
to the formation of aerosols during the times of phytoplankton blooms (Rempillo et al., 2011;
Chang et al., 2011; Park et al., 2017; Sanchez et al., 2018; Abbatt et al., 2019; Quinn et al., 2019).
As well, modelling studies have supported a role for DMS, linked to phytoplankton blooms, as a
contributor to CCN number concentrations in the North Atlantic and Arctic MBLs (Woodhouse et
al., 2013; Zheng et al., 2018; Ghahremaninezhad et al., 2019; Mahmood et al., 2019) and Southern
Ocean MBL (Korhonen et al., 2008; McCoy et al., 2015; Revell et al., 2019). However, the extent
to which DMS can act as a climate regulator remains unclear (Schwinger et al., 2017; Fiddes et
al., 2018), and this role has been refuted (Quinn and Bates, 2011). Analysis of in situ observations
of DMS and its products across the seasonal cycle of marine biogenic activity and in various ocean
regions is needed to improve understanding related to the role of DMS in Earth's climate system.

Marine secondary organic aerosol (SOA) is another important contributor to sub-micron diameter
marine aerosols, but is not well characterized (Rinaldi et al., 2010).  The oceans are a source of a
variety of organic vapors that could lead to SOA formation (O'Dowd and de Leeuw, 2007; Yassaa
et al., 2008; Carpenter et al., 2012; Lana et al. 2012b; Hu et al., 2013; Carpenter and Nightingale,
2015; Kim et al., 2017; Rodríguez-Ros et al., 2020a). Oxygenated volatile organic compounds
(OVOCs) recently linked to photochemical oxidative processes at the sea surface microlayer are
possible contributors to marine SOA (Mungall et al., 2017). Isoprene and monoterpenes appear to
make relatively minor contributions to marine SOA by mass, less than 1% for particles with
diameters smaller than 10 µm at Cape Grim (Cui et al., 2019). The global, annual source of organic
vapors from the oceans is highly uncertain, but current estimates are about 23 to 92 Tg C yr$^{-1}$
(Brüggemann et al., 2018). Laboratory studies indicate that emissions of marine organic vapors
increase with both temperature and incident radiation for temperatures up to about 26 °C
(Meskhidze et al., 2015). Recent observations and modeling studies support a role for Arctic
marine secondary organic aerosol (AMSOA) as a contributor to particle growth to CCN sizes
(Burkart et al., 2017a; Collins et al., 2017; Willis et al, 2017; Willis et al., 2018; Tremblay et al.,
2018; Leaitch et al., 2018;  Croft et al., 2019; Abbatt et al., 2019). For the North Atlantic, organics
are also found to make a large contribution to particle growth to CCN sizes (Sanchez et al., 2018;
Zheng et al., 2020a). The result of the above-noted processes is a large and complex pool of organic
aerosol in the marine environment with sources that vary seasonally and regionally (Cavalli et al.,
2004; Decesari et al., 2011; Cravigan et al., 2015; Liu et al., 2018; Leaitch et al., 2018).

Anthropogenic activity is also an important source of aerosols over the portions of the Earth's
oceans. For the North Atlantic, several previous studies (e.g., Savoie et al., 2002; Stohl et al., 2003;
Huntrieser et al., 2005; Fast et al., 2016) found a key role for synoptic scale motions in lifting
aerosols arising from North American continental emissions and transporting them in layers over
the North Atlantic with intrusions into the MBL. As well, ship traffic is an important source of
both particles and oxidants in the MBL (Corbett et al., 2007; Zanatta et al., 2019; Bilsback et al.,
in press). Ship emissions of nitrogen oxides have a significant control on levels of oxidants such
as ozone, the hydroxyl radical (OH) and $NO_3$ in the MBL (Vinken et al., 2011; Holmes et al.,
2014). In the remote MBL, both OH and $NO_3$ are key oxidants of DMS, along with natural-source
halogens such as BrO, with an important role for multiphase chemistry (Chen et al., 2018).
Interpretation of aerosol observations across several seasons is needed to better understand the
relative contribution of ship emissions to marine particles in the Northwest Atlantic region.

In this study, as part of the Ocean Frontier Institute (www.oceanfrontierinstitute.com), we address
the knowledge gaps that were identified above, concerning several key factors shaping Northwest
Atlantic MBL aerosol size distributions and their seasonal cycle. We consider the role of (1) new
particle formation in clean atmospheric layers near and above the MBL top, (2) particle growth by
marine SOA (MSOA) on descent into the MBL, (3) DMS contributions, (4) ship traffic emissions
and (5) primary sea spray emissions. Aerosol measurements from the North Atlantic Aerosols and
Marine Ecosystems Study (NAAMES) (Behrenfeld et al., 2019) provide an excellent basis for
addressing the role of these five factors in the Northwest Atlantic Ocean region. The NAAMES
aircraft and ship campaigns were conducted during four phases of the Northwest Atlantic annual
plankton cycle from 2015-2018. We interpret the NAAMES aerosol measurements using a state-
of-the-science size-resolved global aerosol microphysics model, GEOS-Chem-TOMAS
(www.geos-chem.org). Our synergistic approach in bringing together NAAMES measurements
and size-resolved aerosol process modeling enables a unique consideration of several key factors
shaping Northwest Atlantic MBL aerosol size distributions and their annual cycle. We also
quantify the impact of these factors on aerosol radiative effects over the North Atlantic.

The second section provides an overview of our measurement and modeling methodology. The
third section presents results using the GEOS-Chem-TOMAS model to interpret NAAMES
aerosol measurements and their seasonal cycle with a focus on the roles of near and above MBL
top NPF, MSOA, DMS, sea spray, and ship emissions. We also quantify the direct and cloud-
albedo indirect aerosol radiative effects attributed to each of these factors during the seasonal
cycle. The final section gives our summary and outlook.

**2. Methodology**

**2.1 Aerosol measurements during the NAAMES campaigns**

NAAMES campaigns were conducted during four key periods in the annual cycle of marine
biogenic activity, namely: the winter transition (November 2015), the accumulating phase
(March/April 2018), the climax transition (May/June 2016), and the declining phase
(August/September 2017) (Behrenfeld et al., 2019). These periods are defined by shifts in net
phytoplankton growth rates and span a wide range in phytoplankton biomass, here estimated from
chlorophyll-*a* concentrations (Chl-*a*). The winter transition is characterized by the annual
minimum in Chl-*a* concentrations (generally < 1 mg m$^{-3}$) and a shift to favor phytoplankton growth
over loss as the increasing ocean mixed-layer depth leads to fewer encounters between
phytoplankton and their grazers. The accumulation phase occurs in early springtime when
increasing sunlight and decreasing ocean mixed layer depths promote increasing phytoplankton
growth rates and concentrations (Chl-*a* between 1 and 2 mg m$^{-3}$). The climax transition is the time
of the annual maximum in phytoplankton biomass (Chl-*a* between 2 and 9 mg m$^{-3}$) and marks the
shift from positive to negative growth rates owing to high grazing rates and depletion of nutrients.
The declining phase (Chl-*a* between 1 and 2 mg m$^{-3}$) occurs later in the summertime when the
ocean mixed layer depth increases and incident sunlight decreases, leading to further declines in
phytoplankton growth and concentrations. Behrenfeld et al. (2019) provide an overview of the four
measurement campaigns, and further details about Chl-*a* during NAAMES. The R/V Atlantis
cruise tracks and NASA C130 flight paths are shown in Figure 1. Due to aircraft mechanical
problems, there were no flights in 2018 during the accumulating phase.

In this study, we examine the NAAMES size-resolved aerosol measurements (particle diameters
20 to 500 nm) from the Scanning Electrical Mobility Sizer (SEMS, Model 138, 2002, BMI,
Hayward, CA) aboard the R/V Atlantis ship. Aerosol particles were isokinetically drawn through
an inlet positioned 18 m above sea level (Bates et al. 2002) and were subsequently dried below
20% relative humidity using silica diffusion driers prior to sampling by the SEMS. Clean marine
periods were identified with criteria of relative wind directions within 90º of the bow, condensation
nuclei number concentrations less than 2000 $cm^{-3}$, ammonium and organic aerosol not covarying,
ammonium $< 100$ ng $m^{-3}$ and having back trajectories primarily over the ocean surface. We also
consider aerosol size-resolved measurements (particle diameters 10 to 282 nm) from the Scanning
Mobility Particle Sizer (SMPS, TSI Inc., Shoreview, MN) aboard the C130 aircraft. As well, we
give attention to measurements of total particle number concentration from the Condensation
Particle Counters (CPCs) with differing nominal lower detection diameters: 3 nm for the CPC
3025 (yielding N3 measurements) and 10 nm for the CPC 3772 (TSI Inc., St. Paul, MN) (yielding
N10 measurements) aboard the C130 aircraft. We also consider submicron, non-refractory sulfate
($SO_4^=$) and organic mass (OM) concentrations from an Aerodyne High Resolution Time-of-Flight
Aerosol Mass Spectrometer (HR-ToF-AMS, DeCarlo et al., 2006) and refractory black carbon
from the Single Particle Soot Photometer (SP2, Schwarz et al., 2006) aboard the aircraft. HR-ToF-
AMS and SP2 measurements are restricted to accumulation-mode aerosol (60-600 nm and 105-
600 nm diameter, respectively). All aircraft observations are made behind a forward-facing,
shrouded, solid diffuser inlet that efficiently transmits particles with aerodynamic diameter less
than 5.0 µm to cabin-mounted instrumentation (McNaughton et al., 2007). Cloud-contaminated
aerosol observations have been removed using a combination of wing-mounted cloud probe and
relative humidity measurements. This filtering may possibly obscure some NPF events in
proximity to clouds and remove some cloud-processed samples from the vertical profiles. Aerosol
number and mass concentrations are reported at standard temperature and pressure. A Proton-
Transfer-Reaction Time-of-Flight Mass Spectrometer (PTR-ToF-MS) (Müller et al, 2014;
Schiller, 2018) was used aboard the NASA C-130 to measure volatile organic compounds
including DMS and acetonitrile. Both observational and model data for periods where acetonitrile
concentrations exceed 200 ppt are filtered out following Singh et al. (2012) to remove significant
biomass burning contributions that are not the focus of this study.

## 2.2 GEOS-Chem-TOMAS model description

We use the GEOS-Chem model (v12.1.1) (http://www.geos-chem.org) coupled to the TwO Moment Aerosol Sectional (TOMAS) microphysics scheme (Adams and Seinfeld, 2002; Lee and Adams, 2012; Kodros and Pierce, 2017), with 15 sections, representing particle sizes from 3 nm to 10 μm. All simulations are at a 4° × 5° resolution with 47 vertical levels extending to 0.01 hPa. The meteorological fields are from the GEOS Forward Processing off-line fields (GEOS-FP; https://gmao.gsfc.nasa.gov/GMAO_products/). Our size-resolved aerosol simulations parameterize the processes of particle nucleation, coagulation, condensation, along with wet and dry deposition and include the in-cloud aerosol coagulation scheme of Pierce et al. (2015). Sulfate, organic and black carbon, sea salt, dust and aerosol water are simulated. TOMAS is coupled to the full tropospheric aerosol/chemistry scheme of GEOS-Chem. Wet deposition follows Liu et al. (2001), Wang et al. (2011) and Wang et al. (2014). To represent efficient wet removal by North Atlantic drizzle in October and November, we implement a fixed in-cloud removal efficiency of 0.001 s$^{-1}$ in the lowest 2 km of the model atmosphere over the ice-free ocean and enable wet removal of sulfate and organic aerosol in clouds with temperatures between 237 K and 258 K. In all seasons, we use the GEOS-FP cloud fraction as the precipitation fraction in the model layers where precipitation occurs for a closer connection with the meteorological fields (Croft et al., 2016b; Luo et al., 2019; Luo et al., 2020). Dry deposition uses the resistance in series approach of Wesley (1989). Simulated gas-phase species are also removed by dry and wet deposition as described in Amos et al. (2012).

For emissions, we use the GEOS-Chem v 12.1.1 default setup for gas-phase and primary aerosol emissions. We use emissions from the Community Emissions Data System (CEDS) for global anthropogenic sources of NO$_x$, CO, SO$_2$, NH$_3$, non-methane VOCs, black carbon, and organic carbon, including from international shipping as a source of both primary and secondary particles. Primary particles are emitted with a lognormal distribution (Lee et al., 2013). The most recent CEDS emissions dataset extends to the year 2017, as described in McDuffie et al. (2020). In this work, monthly CEDS emission totals for each compound are spatially gridded by source sector, according to the 0.1° × 0.1° gridded EDGAR v4.2 emissions inventory (EC-JRC/PBL, 2012) and population, as described in Hoesly et al. (2018). To account for in-plume chemical processing of

ship emissions, we use the PARANOX scheme of Holmes et al. (2014). CEDS emissions are
overwritten over the United States by the National Emissions Inventory (NEI11) with updated
scale factors for our simulation years (2015-2018). We calculated these factors based on emission
data for these years from the United States Environmental Protection Agency. Over Canada, we
use the Air Pollutant Emissions Inventory (APEI). The Global Fire Emissions Database (GFED4s)
is used for biomass burning emissions (van der Werf et al., 2017) for the years 2015-2016, with
GFED4s climatological values for 2017 and 2018 since exact-year emissions were not available
when we conducted our simulations. Dust emissions are from the scheme of Zender et al. (2003).
Sea salt emissions follow Jaeglé et al. (2011). This temperature-dependent parameterization
decreases global emissions relative to the Gong (2003) parameterization. A coupled
parameterization for primary organic aerosol from sea spray was not available for our aerosol size-
resolved GEOS-Chem-TOMAS simulations, such that some sea spray organics could be
misrepresented as sea salt, since all sea spray in our simulations is considered sea salt. Such
primary organic emissions are expected to have no seasonal cycle when averaged over the
NAAMES region (Bates et al., 2020).

Exchange of DMS between the ocean and atmosphere is parameterized using the default GEOS-
Chem parameterization, which follows Johnson (2010), largely based on Nightingale et al. (2000a;
2000b). We use the 8-day mean satellite-retrieval seawater DMS dataset of Galí et al. (2019)
developed using the methodology of Galí et al. (2018), for available years (2015 and 2016) for the
region north of about 40 °N. The Lana et al. (2011) DMS climatology is used elsewhere. Terrestrial
biogenic emissions are from MEGAN2.1 as described in Guenther et al. (2012). Following Croft
et al. (2019), we add a source of MSOA coupled to the simple SOA scheme described in Pai et al.
(2020). Emissions of MSOA-precursor vapors have been found to increase with temperature
(Meskhidze et al., 2015; Rodríguez-Ros et al., 2020a; Rodríguez-Ros et al., 2020b). Here, we use
a temperature-dependent simulated source of MSOA-precursor emissions ($S_{MSOA}$), $S_{MSOA}$ = 70T
+ 350 µg m$^{-2}$ d$^{-1}$, where T is atmospheric temperature (°C) at 2 m altitude. The values of 70 and
350 are found to yield acceptable model-measurement agreement for NAAMES campaign-median
ship-track and aircraft measurements (Supplementary Figs. S1-S4 and Supplementary Tables S1
and S2). This simulated source of condensable vapors is emitted with a 50/50 split between vapors
that are immediately available to form MSOA and vapors with 1-day aging prior to availability
(and not susceptible to wet removal). MSOA contributes to particle growth in our simulations (in
agreement with observational-based studies e.g., Sanchez et al., 2018; Zheng et al., 2020a), along
with sulfuric acid, but since the particle nucleating abilities of MSOA are unclear, it does not
contribute to new-particle formation.

All simulations include particle nucleation in the boundary layer that is parameterized with the
ternary ($H_2SO_4$-$NH_3$-$H_2O$) scheme of Napari et al. (2002), which was scaled by $10^{-5}$ to better match
continental boundary-layer measurements (Westervelt et al., 2013). The binary ($H_2SO_4$-$H_2O$)
scheme of Vehkamaki et al. (2002) is employed in the free troposphere at low $NH_3$ concentrations.
Growth and loss of particles smaller than 3 nm are approximated following Kerminen et al. (2004).
In our simulations, as a surrogate for unparameterized processes in the lower free troposphere and
near the MBL top, we also employ an activation-type nucleation parameterization from the MBL
top to about 2 km altitude. This activation-type scheme parameterizes nucleation rates as a linear
function of sulfuric acid concentrations, using an empirical factor ($A = 2 \times 10^{-6}$ s$^{-1}$) (Kulmala et
al., 2006; Sihto et al., 2006), and serves as a proxy representing several unknown/unparameterized
mechanisms related to NPF. Pockets of very clean air with low condensation sink near MBL
clouds, which favor new particle formation (Kazil et al., 2011), are not resolved by large-scale
models such as ours, with grid boxes on the scale of 100s km$^2$. Efficient wet removal by drizzling
MBL clouds contributes to these pristine conditions (Wood et al., 2017). As well, MBL clouds
reflect ultraviolet (UV) radiation and create pockets of enhanced UV, which favors photochemical
production of aerosol precursor vapors (Weber et al., 2001; Wehner et al., 2015), that are not
resolved by our model. Additionally, the particle nucleating capacity of MSOA is unclear and
particle formation parameterizations are not yet developed to represent NPF when several gas-
phase precursors interact. These precursors include, but are not limited to, MSA (Chen et al.,
2016), HPMTF (Veres et al., 2020), amines (Facchini et al., 2008), iodine (Allan et al., 2015), and
other extremely low-volatility organic compounds (ELVOCs) (Riccobono et al., 2014). The extra
nucleation in the lower troposphere with the activation-type parameterization represents particle
precursors that could have the same source as sulfuric acid. This approach may not capture the
timing and magnitude of the variability in NPF correctly because the vapors participating in this
nucleation are likely not just sulfuric acid. Future work is needed to better understand the nature
of the nucleating species in the lower troposphere over the oceans.

We also conduct off-line radiative transfer calculations using the Rapid Radiative Transfer Model
for Global Climate Models (RRTMG) (Iacono et al., 2008) to assess the direct radiative effect
(DRE) and cloud-albedo aerosol indirect effect (AIE). The aerosol optical properties are calculated
using the Mie code of Bohren and Hoffman (1983) to find the extinction efficiency, single
scattering albedo, and asymmetry factor. Then, these optical properties, along with the monthly
mean cloud fraction and surface albedo from the GEOS-FP meteorology fields, are input to the
RRTMG to determine the change in top-of-the-atmosphere solar flux (DRE) between two
simulations (our control simulation and one of the sensitivity simulations, Sect. 2.3). Our DRE
calculations follow Kodros et al. (2016), with updates to include ammonium nitrate as described
in Bilsback et al. (in press). All particles except black carbon are treated as internally mixed within
each size section. We also calculate the cloud-albedo aerosol indirect effect (AIE) as described in
Kodros et al. (2016), Croft et al. (2016a) and Ramnarine et al. (2019). The Abdul-Razzak and
Ghan (2002) parameterization is used to calculate offline cloud droplet number concentrations
(CDNC) using the aerosol mass and number concentrations from our simulations. We assume an
updraft velocity of 0.5 m s$^{-1}$ and the hygroscopicity parameters used by Kodros et al. (2016) and
Kodros and Pierce (2017), assuming aerosol internal mixture, including ammonium nitrate
following Bilsback et al. (in press). For each model grid box, we assume cloud droplet radii of 10
μm and perturb this value with the ratio of the monthly mean CDNC between two simulations (our
control simulation and one of the sensitivity simulations, Sect. 2.3), assuming constant cloud liquid
water content. The RRTMG is used to calculate the change in the top-of-the-atmosphere solar flux
(AIE) due to changes in cloud droplet radii.

As one evaluation of simulation performance, we calculate the mean fractional error (MFE) of the
0[th] to 3[rd] moments between the simulated and observed MBL aerosol size distributions, following
Boylan and Russell (2006) and using the same methodology as Hodshire et al. (2019) and Croft et
al., (2019). The MFE is defined as a mean over the *N* aerosol size distribution moments,

$\text{MFE} = \frac{1}{N} \sum_{i=0}^{i=N-1} \frac{abs|C_m(i) - C_o(i)|}{(C_m(i) + C_o(i))/2}$             (1)

where $C_m(i)$ is the integrated value of the $i^{th}$ moment of the simulated aerosol size distribution and
$C_o(i)$ is the integrated value of the $i^{th}$ moment of the observed aerosol size distribution. The MFE
can range from 0 to +2. We adopt the convention of Boylan and Russell (2006) to consider a MFE
of 0.5 or less as acceptable.

For consideration of vertical profiles, we binned the measurement and simulation values using a
500 m height resolution, starting from the surface to 500 m as the first bin. Campaign-median
values are calculated within each bin and plotted at the mid-point of the bin, starting at 250 m.
During NAAMES, the lowest aircraft flight level altitude was around 150-200 m GPS altitude. We
use a plane-flight diagnostic in the model to sample the simulation interpolated between grid-cell
centers to the aircraft-flight-track position, during the times when measurement data was available
for each respective instrument. We find consistent results with bin resolutions of 250, 500 and
1000 m, giving support for our selected binning resolution. The vertical profiles show
measurements and model output along the aircraft flight tracks only and do not include any
measurements or model output for the ship track. Vertical profile MFEs (Eq. 1) are calculated by
summation over the altitude bins.

**2.3 Summary of GEOS-Chem-TOMAS simulations**

Table 1 summarizes the simulations conducted. Simulation BASE is our control simulation and
includes all emissions and process parameterizations described above. We conduct five sensitivity
simulations to examine the role of several key factors involved in shaping the aerosol distributions
within the NAAMES study region. Simulation noABLNUC is the same as BASE, except without
the sulfuric acid-dependent activation-type surrogate nucleation parameterization, which we
implemented from the MBL top to about 2 km. Simulation noMSOA is the same as BASE, but
without the source of temperature-dependent condensable marine organic vapors, forming MSOA.
Simulation noDMS is the same as BASE, but without DMS. Simulation noSHIPS is the same as
BASE, but without any ship emissions. Finally, simulation noSS is the same as BASE, but without
any primary sea spray emissions. All simulations are sampled coincidentally with the
measurements using hourly output along the NAAMES aircraft and ship tracks within the
respective model grid boxes, using the NAAMES campaigns' 1-minute-resolution navigation data.
To manage computational expense, the simulations are necessarily at a coarse resolution, which
can bias model-measurement comparisons. However, these biases will be lower for remote marine
regions such as the NAAMES study region than over land regions, which generally have greater
spatial inhomogeneity. Representativeness errors were also reduced by limiting our model-
measurement comparisons to campaign-median values.

## 3. Results and Discussion


### 3.1 Key features of aerosols observed during NAAMES


Aerosol observations made during the NAAMES campaigns were in four seasons, capturing
different stages of the annual cycle of Northwest Atlantic marine biogenic activity (Behrenfeld et
al., 2019). Figure 2 shows the campaign-median marine-influenced aerosol size distributions from
SEMS (particle diameters 20-500 nm) for the four R/V Atlantis cruises. November 2015 (winter
transition, bloom minima) is characterized by the lowest aerosol number concentrations. The peak
of the Northwest Atlantic drizzle season occurs at this time, with efficient wet removal of
accumulation-sized aerosol (diameters larger than about 50 to 100 nm) (Browse et al., 2012). As
well, relative to other the seasons, marine biogenic emissions are low at this time of minimal
phytoplankton biomass. The summertime observations during both May/June 2016 (climax
transition, phytoplankton bloom maxima) and August/September 2017 (declining phase) are
characterized by a weakly dominant Aitken mode (particle diameters < 100 nm). The winter
transition (November 2015) and early spring accumulation phase observations (March/April 2018)
are characterized by the dominance of accumulation-mode aerosols (particle diameters > 100 nm).

The vertical profiles of campaign-median integrated-SMPS (particle diameters of 10 to 282 nm)
observations are shown in Fig. 3. There are several key features of the observed aerosol vertical
profiles for the three NAAMES flight campaigns. These profiles exhibit several particle number
maxima in the lower free troposphere below 6 km, including below 2 km during the May/June
climax transition period. As shown in Fig. 3, aerosol surface area and volume are less at altitudes
below about 3 km relative to altitudes above 3 km. This lower particle surface area at these altitudes
favors NPF over growth of pre-existing particles as available vapors condense in these relatively
cleaner atmospheric layers (Kazil et al., 2011). Transport of aerosols (in part associated with
continental emissions) contributes to particles in all seasons. Fast et al. (2016) characterized
summertime North Atlantic transport layers in the free troposphere associated with synoptic-scale
lifting. The late fall (November 2015, Fig.3) is characterized by the lowest aerosol number, surface
and volume concentrations, similar to the findings shown in Fig. 2.

Figure 4 shows the vertical-profile campaign-median total particle number concentrations from
CPCs, for aerosols with diameters larger than 3 nm (N3), larger than 10 nm (N10), and the
difference between the two (N3-N10). For the May/June 2016 climax transition (phytoplankton
bloom maximum), there are enhancements in observed number concentration (N3, N10 and N3-
N10) below about 2 km in the free troposphere, indicating NPF at these altitudes (Fig. 4). The
MBL top ranged from about 0.5 to 2 km for the NAAMES cruises (Behrenfeld et al., 2019). The
lower free tropospheric region near and above the MBL top is an important region for marine
NPF. These altitudes above the MBL clouds are generally very clean, which favors NPF, and
strongly sunlit, which favors the photochemical oxidative production of particle precursors for
NPF. Previous studies based on observations from other marine regions have also found a cloud-
processed ultra-clean layer with weak condensation/coagulation sinks at about 1 km altitude,
where NPF is favored (Kazil et al., 2011; Takegawa et al., 2020). Figure 4 also shows
enhancements in the observed N3 and N10 concentrations below 6 km during the declining
phase and winter transition (bloom minima). However, the total number concentration
enhancements below 2 km are most pronounced during the phytoplankton bloom maximum,
suggesting a connection between particle number and the level of marine biogenic activity.

$SO_4^=$ and OM are dominant non-refractory components of the submicron-diameter aerosols, and
vertical profiles of campaign-median observations are shown on Fig. 5. During the summertime
(May/June 2016, climax transition and August/September 2017, declining phase), the OM
contribution exceeds that of $SO_4^=$ at most altitudes up to 6 km. Non-refractory $SO_4^=$ has its peak
contribution during the climax transition season. This May/June phytoplankton bloom maxima
period is the time of peak observed near-surface atmospheric DMS mixing ratios, as shown in Fig.
6. During the climax transition (bloom maxima), non-refractory $SO_4^=$ concentrations increase
towards the surface, suggesting a marine surface source, similar to summertime Arctic marine
profile observations (Willis et al., 2017). Black carbon (BC) concentrations are also shown in Fig.
5 and have several peaks in the free troposphere in all seasons, consistent with a long-range
transport source. Maximum BC concentrations are in May/June, likely associated with greater
transport of anthropogenic continental pollution and biomass burning during this time, relative to
other seasons. Springtime has also been associated with peak BC concentrations in the Arctic due
to long-range transport (Sharma et al., 2004; Sharma et al., 2006; Fisher et al., 2010; Wang et al.,
2011; Xu et al., 2017). All aerosol mass concentrations in the lowest 2 km of the atmosphere (Fig.
5) are lowest in the November 2015 winter transition, which is a time of efficient wet removal by
drizzle (Browse et al., 2012; Wood et al., 2017), diminishing marine emissions due to diminishing
phytoplankton biomass, and outbreaks of relatively less polluted polar air advected down the
Labrador Strait (Behrenfeld et al., 2019). For the Arctic, the fall season has also been associated
with a relative minimum in aerosol number concentrations (Tunved et al., 2013; Croft et al.,
2016b).

The GEOS-Chem-TOMAS model (described in Sect. 2.2 and 2.3) is generally able to simulate the
above-noted features of the aerosols over the Northwest Atlantic. Simulation BASE captures key
aspects of the MBL size distributions including the minimum in aerosol number during the
November winter transition, the weakly dominant Aitken mode during the May/June climax
transition and August/September declining phase and the maximum in number of accumulation-
mode particles (diameters greater than 100 nm) during the March/April accumulation phase,
despite errors such as between 20-50 nm (Fig. 2). As well, the BASE simulation captures several
lower tropospheric enhancements in particle number concentration, although the simulated altitude
for the maximum is sometimes displaced and there are errors in the magnitude (Figs. 3 and 4). In
the lowest 2 km of the atmosphere, $SO_4^=$, OM, and BC mass concentrations for simulation BASE
are generally within the 25[th] to 75[th] measurement percentiles, except for BC and OM
underpredictions in May/June 2016, and OM overprediction in November 2015. All simulated
$SO_4^=$ presented in this study is non-sea-salt $SO_4^=$. Simulation BASE also captures that the near-
surface $SO_4^=$ is greatest during the May/June climax transition and the near-surface OM has its
maximum value during the August/September declining phase. For each season the mean MFE
across the parameters considered in Figs. 2 to 5 (BASE versus measurements, Supplementary
Table S2) is satisfactory (MFE ranges 0.43 to 0.50). In the next four sub-sections, we use the
GEOS-Chem-TOMAS BASE simulation, relative to a set of sensitivity simulations, to examine
the potential of five key factors to shape aerosol size distributions in the Northwest Atlantic during
four stages of the annual cycle of marine biogenic activity.

**3.2 Role of new particle formation (NPF) in the lower troposphere**

Our simulations (BASE relative to noABLNUC, Fig. 4) suggest that NPF near and above the MBL
has a strong control on the development of the total particle number (N3) maxima, with peak
magnitude during the phytoplankton bloom maxima in layers below 2 km. Without the surrogate
NPF scheme employed near and above the MBL top, the ternary NPF scheme in the MBL in
simulation noABLNUC fails to simulate sufficient particle number, although vertical-profile
campaign-median ammonium concentrations below 4 km altitude had acceptable agreement with
observations (MFE ranges from 0.12 to 0.48, not shown). Figure 4 shows about a one-order-of-
magnitude underprediction of N3 below about 2 km for noABLNUC. NoABLNUC has an
unacceptable seasonal-mean model-measurement agreement across the measurement set (MFE
ranges from 0.66 to 0.78, Supplementary Table S2). Figure 3 also shows that NPF near and above
the MBL top makes a significant contribution to simulated particle number concentrations for
aerosol diameters of 10 to 282 nm in the lower troposphere, most strongly in the summertime
(BASE relative to noABLNUC). There is little impact on aerosol mass concentrations for
simulation noABLNUC relative to BASE (Fig. 5).

The simulated N3-N10 (Fig. 4) illustrates that representation of NPF is a challenge for models,
because there are difficulties capturing the magnitude and altitudes of the N3-N10 maxima. These
discrepancies reflect key knowledge gaps related to the species that can form new particles in the
marine environment (e.g., Veres et al. 2020). As well, the coefficient that we used for the surrogate
activation-style nucleation parameterization was derived for a continental environment. The
empirical ('A') value used by the parameterization appears to yield excessive NPF for the
NAAMES marine environment. Activation-style nucleation was added in our simulations as a
proxy for missing nucleation when the condensation sink is low, and conditions favor high
oxidation rates. We acknowledge that this approach will miss variability in the timing and rates
because it is a surrogate and not exactly the correct mechanism. As well in the summertime, the
simulations underpredict N3-N10 concentrations above 2 km, suggesting the need for future work
to better understand the NPF processes at these levels, where the binary scheme of Vehkamaki et
al. (2002) does not generate sufficient NPF.

NPF also makes a very strong contribution to the simulated aerosol size distributions within the
MBL near the ocean surface (BASE versus noABLNUC, Fig. 2). Although our simulations do
include NPF within the MBL, simulated NPF occurs more strongly near and above the MBL top
and the resultant particles grow by condensation of available vapors and cloud processing while
descending into the MBL. This role for NPF is in agreement with previous studies including those
of Clarke et al. (2013), Quinn et al. (2017), and Williamson et al. (2019). As a result, NPF from
several altitudes above the ocean surface contributes to the near-ocean-surface particles, with
diameters from 20 to 200 nm. NPF does occur in the MBL. However, those levels above the MBL
clouds favor oxidative chemistry that yields particle precursors, particularly from the wide-spread
and persistent DMS sources in the marine environment (Kazil et al., 2011). Table 2 shows that for
all seasons, the surrogate nucleation (simulation BASE, MFEs ranging from 0.04 to 0.33)
represents an improvement over simulation noABLNUC (without this surrogate NPF
parameterization, MFEs ranging from 0.50 to 0.95).

Extending the surrogate activation-style parameterization to the surface (Supplementary Figs. S5-
S8 and Supplementary Table 3), leads to overprediction of the number of particles with diameters
less than 50 nm in the MBL and yields higher MFEs (ranging from 0.20 to 0.56) than for simulation
BASE, although the errors were not as large as those for noABLNUC. For the vertical profiles,
this extra NPF extended into the MBL yields overprediction of N3, N10, and N3-N10 below 1 km
in all seasons. Aerosol surface area and volume (in the SMPS particle-diameter size range of 10
nm - 282 nm) were also over predicted during the August/September declining phase, when the
simulated temperature-dependent MSOA source was strongest, growing these extra new particles
to larger sizes. These challenges highlight the relevance of ongoing research to better understand
NPF in the marine environment.

**3.3 Role of particle growth by condensing marine organic vapors**

Condensing marine organic vapors forming MSOA are needed in our simulations (in addition to
$H_2SO_4$) for sufficient particle growth to yield satisfactory model-measurement agreement for MBL
size distributions (BASE versus noMSOA, Fig. 2). For simulation noMSOA, the model
overpredicts the number of particles with diameters smaller than about 30 nm in the MBL. Due to
insufficient particle growth of these sub-30 nm particles, the number of particles with diameters
between about 30 to 200 nm is underpredicted by more than 50% for simulation noMSOA.

In our simulations, MSOA enables particle growth to CCN sizes (diameters of about 50 nm or
larger). After particles reach CCN sizes, cloud processing can also contribute to simulated particle
growth towards accumulation-mode particles (diameters of 100-1000 nm) due to aqueous-phase
aerosol production. Other cloud processes include coagulation of cloud droplets with each other
and with interstitial aerosols (Hoose et al., 2008; Pierce et al., 2015). Our simulations include the
latter and aqueous-phase sulfate production. As clouds evaporate, cloud processing leads to
development of the 'Hoppel minima' of the MBL aerosol size distributions (Hoppel et al., 1987),
which is the minimum aerosol diameter that activates to form a cloud droplet (about 50-70 nm for
the observations in Fig. 2). This minimum diameter is smallest in the winter transition (November
2015), suggesting that smaller particles activated under the clean condition of this season relative
to the other seasons. As shown by Table 2, simulation noMSOA has an unacceptable annual-mean
MFE of 0.63, larger than the MFE of 0.23 for simulation BASE, which includes particle growth
due to MSOA.

The nature and flux of marine vapors forming MSOA are not well understood. As a result, we
developed a simplistic MSOA parameterization for use in this study, such that the MSOA
precursors vapor emissions are an increasing function of temperature. This approach yields a
seasonal cycle, and is in agreement with the temperature dependence trend found by previous
studies, including Meskhidze et al. (2015), Rodríguez-Ros et al. (2020a) and Rodríguez-Ros et al.,
2020b). We find that the simulated NAAMES cruise-track median aerosol size distributions are
sensitive to the coefficients used in the parameterization ($S_{MSOA}$ = 70T - 350 μg m$^{-2}$ d$^{-1}$)
(Supplemental Figs. S1 and Table S1). For example, varying the temperature sensitivity between
50-100 and the intercept between 300-500 change the simulated number concentration of particles
with diameters larger than 50 nm in the MBL by up to a factor of two, with the greatest sensitivity
during the summertime (Supplemental Fig. S1). For the NAAMES MBL size distributions, the
annual-mean model-measurement MFEs are acceptable (ranging from 0.23 – 0.38, lowest for
BASE) for all temperature-dependent parameterizations that we tested, except for the factor-of-
ten scaling up of the BASE MSOA parameterization (simulation 10x(70T-350), Supplementary
Table S1, MFE of 0.75) and with the MSOA parameterization removed (simulation noMSOA,
Supplementary Table S1, MFE of 0.63). While this source flux is reasonably constrained for our
simulations, future work is needed to better understand and parameterize this source.

The vertical profiles are also sensitive to the MSOA parameterization (Supplementary Figs. S2-
4). Between noMSOA and the various MSOA parameterizations that we tested, concentrations
vary by up to a factor of about 2 for aerosol number (N3, N10, and N3-N10), SMPS-size-range
(diameters 10 nm - 282 nm) number, surface area, volume and also OM. Simulation noMSOA has
relatively greater error in the mean across the entire measurement set for each season (MFE ranges
from 0.53-0.68) relative to BASE (MFE ranges from 0.42-0.50) (Supplementary Table S2).

Although the chosen MSOA parameterization reasonably represents the observations, major
knowledge gaps remain regarding MSOA precursor species and their chemical lifetimes. While
the nature of MSOA precursors is not well-understood, recent measurements suggest that these
precursors could include a variety of chemical compounds. For example, measurements from the
Arctic indicate that the organics in marine aerosols were not typical biogenic SOA but had a long-
hydrocarbon chain implying a fatty acid type precursor (Willis et al., 2017). In other marine
regions, isoprene (Ciuraru et al., 2015) and carboxylic acids (Chiu et al., 2017) may also be
important. Given the limitations of current knowledge and the indications for a variety of MSOA
precursors, the improved MFEs for BASE relative to noMSOA provide support for the employed
MSOA parameterization.

The near-surface campaign-median climax transition and declining phase OM concentrations are
within the 25th to 75th measurement percentiles for simulation BASE, and below the 25th percentile
of the observations for simulation noMSOA (Fig. 5). On average over the lowest 2 km of the
atmosphere during the May/June climax transition and August/September declining phase,
simulation BASE relative to noMSOA indicates that MSOA contributes about 200-400 ng m$^{-3}$ to
simulated OM. Saliba et al. (2020) suggest that MBL-measurement non-refractory OM during
NAAMES clean marine periods provides a good estimate of MSOA. Their seasonal-average non-
refractory OM of about 300-400 ng m$^{-3}$ for the 2016 May/June climax transition (phytoplankton
bloom maxima) and 2017 August/September declining phase is similar to our model result. This
contribution is about 3- to 4-fold greater than the contribution upwards of 100 ng m$^{-3}$ from previous
studies, noted in Kim et al. (2017). The model-measurement agreement for OM for 2017 is
influenced by significant biomass burning with high altitude emission injections during this time
(Zheng et al., 2020b; Saliba et al., 2020). Errors in the simulated emissions due to use of a GFED
climatological-year emissions and injection-height errors could account for some of the model-
measurement bias at high altitudes. As well, despite our implementation of a filter to remove
measurement and model samples with strong in-plume aerosol enhancements during times of high
acetonitrile concentrations, some biomass burning influence still affects the presented vertical
profiles. Below 500 m altitude, condensing organic vapors yielding MSOA also increase the
simulated aerosol surface area and volume by a factor of about 2-3 in all seasons (noMSOA versus
BASE, Fig. 3), to be slightly over the 75$^{th}$ percentile of the observations (Fig. 3). Surface area and
volume results from the simulation are very sensitive to the size-distribution simulation near the
282 nm diameter cut-off that contributes to differences between these simulations.

Figure 4 demonstrates that MSOA has a feedback on NPF. With lower aerosol surface area and
lower condensation sink (noMSOA), the N3 and N3-N10 below 2 km altitude are strongly
overpredicted because NPF increases and a lack of growth to larger sizes impacts N3-N10. During
November, the N3 and N3-N10 overprediction also occurs at altitudes above 2 km because MSOA
has a relatively greater influence on aerosol surface area at those altitudes in this season (Fig. 3).
In this less-polluted late fall season, the influence of MSOA is relatively stronger at higher altitudes
than in other seasons. Model-measurement agreement improves for N3 and N3-N10 with the
addition of MSOA (simulation BASE relative to noMSOA, Fig. 4). Kazil et al. (2011) also found
that condensing vapors generate a condensation sink that moderates the level of NPF in the marine
environment. As well, recent studies from the Arctic indicate a key contribution to particles from
condensing marine organic vapors (Burkart et al., 2017a; Willis et al., 2017; Croft et al., 2019).
The impact of MSOA on the simulated N10 vertical profiles is small. The cloud filtering, which
we applied to the model and measurement aerosol samples along the flight track, preferentially
removes some of the cloud-processed samples, and contributes to this result.

**3.4 Role of DMS**

Figure 2 shows that DMS also has a control on the simulated MBL aerosol size distributions
(BASE versus noDMS) for the four seasons of the NAAMES campaigns. The total simulated
number of particles attributed to DMS is lowest during the phytoplankton bloom minima (winter,
November 2015) and greater in other seasons. For example, for particle diameters at 40 nm, the
DMS-related contribution to the size distribution (Fig. 2) is about 200-300 $cm^{-3}$ in all seasons,
except less than 50 $cm^{-3}$ during the bloom minima. Sulfuric acid from the oxidation of DMS has a
two-fold role in both NPF and in growing particles. However, as indicated by simulations
noABLNUC and noMSOA relative to BASE (Fig. 2), the DMS contribution is in concert with
both (1) a source of condensable marine organic vapors and (2) NPF near and above the MBL top.
The contribution of DMS to MBL particles is consistent with the findings of many previous
studies, including Chang et al. (2011), Ghahremaninezhad et al. (2016), Park et al. (2018), Sanchez
et al. (2018), Mahmood et al. (2019), Quinn et al. (2019) and Veres et al. (2020).

Simulation noABLNUC relative to noDMS for the marine-influenced MBL size distributions (Fig.
2) suggests that anthropogenic influences make a contribution as a source of particle-precursor
vapors for NPF, in addition to DMS. This relative contribution is particularly strong during the
accumulation phase (March/April 2018). In our simulations, anthropogenic $SO_2$ is oxidized to
$H_2SO_4$ and contributes to the particle precursors for NPF near and above the MBL top (in addition
to DMS oxidation products), followed by particle growth on descent into the MBL. As a result,
Fig. 2 shows a greater underprediction of aerosol number for simulation noABLNUC versus
noDMS.

Figure 6 indicates that the simulated DMS is generally consistent (MFEs ranging from 0.12 to
0.26, Supplementary Table S2) with the observed DMS mixing ratio vertical profiles and their
seasonal cycle for the NAAMES campaigns. DMS makes the strongest contribution to simulated
sulfate mass concentrations in the lowest 2 km during the May/June climax transition, reducing
model-measurement bias from about 40% to 10% (Fig. 5). Figures 3 and 4 suggest that in the
lowest 2 km of the atmosphere, DMS contributes to both NPF and particle growth as there are
increases in N3, N10, N3-N10, particle surface area and volume for simulations BASE versus
noDMS. However, this behavior is co-dependent on conditions favorable to NPF near the MBL
top and a source of MSOA.

**3.5 Role of ship traffic emissions**

Ship emissions are a source of primary and secondary particles, as well as a control on oxidants
(Corbett et al., 2010; Vinken et al., 2011; Holmes et al., 2014). Our simulations suggest that ship
emissions are also a control on the NAAMES-region MBL marine-influenced aerosol size
distributions (Fig. 2, noSHIPS versus BASE). For example, for the simulated summertime MBL
size distribution at particle diameters at 40 nm, about 100-200 particles $cm^{-3}$ are attributed to ship
emissions (Fig. 2). Table 2 shows that during the phytoplankton bloom and March/April
accumulating phase, the noSHIPS simulation agrees more closely with the measurements than the
BASE simulation, although both are within acceptable agreement (MFE < 0.5). These simulation
challenges highlight the importance of future work to better understand the role of oxidants from
ship emissions on particle production in the marine environment and to understand the size
distribution of primary marine emissions.

Ship emissions make about a 50% contribution to the simulated sulfate campaign-median near-
surface mass concentration in vertical profiles over the NAAMES study region (Fig. 5). For our
simulations the impact of ship emissions on particle number is mostly limited to the lowest 2 km.
Simulation BASE relative to noSHIPS suggest that about 10% of the N10 in the lowest 500 m of
the atmosphere is attributed to ship emissions (Fig. 4). Figure 4 (right column) indicates that among
the five factors considered by our sensitivity studies, ship emissions are among the smallest
influence on the NPF. Major trans-Atlantic ship traffic routes (Corbett et al. (2007) are included
in the NAAMES study region. Enhancements in observed benzene mixing ratios in the MBL
relative to other long-lived tracers of anthropogenic emissions such as acetone (not associated with
ship traffic) are observational evidence that ship emissions influence the study region
(Supplementary Fig. S9).

Figure 6 demonstrates that atmospheric DMS mixing ratios are also sensitive to ship emissions.
This effect occurs because ship emissions are a control on oxidants in the MBL, and enhance OH
and NO$_3$, which are chemical sinks of DMS. As a result, simulated DMS mixing ratios increase
when ship emissions are removed. As ship traffic is expected to change in future years with
changes to routes and regulations (Gilgen et al., 2018; Bilsback et al. (in press)), the relative
importance of ship emissions in the North Atlantic MBL will likely change.

**3.6 Role of sea spray**

Figure 2 shows that simulated sea spray acts as a condensation sink in the MBL. Without sea spray
emissions, there is an increase in the number of sub-200 nm diameter particles (simulation noSS
relative to BASE). However, this relative increase in simulated number is less than that attributed
to other factors considered in the previous sections. While not a strong contributor to particle
number in our simulations, sea spray is the dominant contributor to aerosol mass.

The simulated campaign-median MBL sea spray mass concentrations are within the measurement
range of 3-8 µg m$^{-3}$ found by Saliba et al. (2019) (Supplementary Fig. S10), despite the
considerable uncertainties related to size-resolved sea spray emissions (e.g., Bian et al., 2019;
Regayre et al. (2020)). Regayre et al. (2020) found that global sea spray emissions could be under
predicted by a factor of 3 by the Gong (2003) parameterization. We conducted a simulation with
factor-of-3 scaling of the sea spray emissions (Supplementary Figs. S11-S14, Supplementary
Table S4) and found a decrease in MBL number concentrations, rather than an increase. This
reduction occurred because the enhanced condensation sink from the additional sea spray
emissions suppressed NPF. Our simulations use the Gong (2003) parameterization with a sea-
surface-temperature-based scaling as described by Jaeglé et al. (2011), so are not directly
comparable to the Regayre et al. (2020) findings. Nonetheless, these findings highlight the
importance of ongoing work to improve size-resolved sea spray emissions parameterizations in
models. The direct radiative effect of this sea spray mass loading is examined in the following
section.

**3.7 Radiative effects attributed to NPF near MBL top, MSOA, DMS and ship emissions**

Figure 7 shows the simulated two-month mean direct radiative effect (DRE) attributed to the five factors we consider, (1) NPF near and above the MBL top, (2) MSOA, (3) DMS, (4) ship emissions and (5) primary sea spray emissions and magnitude of the regional-mean DREs over a region of the North Atlantic (40-60 °N, 20-50 °W). We note that the radiative effects attributed to the separate factors are not linearly additive because the factors impact each other non-linearly. For example, turning off either MSOA or nucleation above the boundary layer would shut down the majority of the production of accumulation-mode particles in the MBL (Fig. 2) since these particles require both nucleation and growth. Hence, adding the radiative effects from these two factors would result in double counting some radiative effects. Figure 7 indicates that the strongest calculated DRE is attributed to sea spray, which dominates the aerosol mass loading in the MBL. The sea spray DRE has a maximum during the 2018 March/April accumulating phase, which is a time of frequent synoptic scale storms with strong winds. Stormy conditions prevented the R/V Atlantis from travelling north of 45 °N during this final NAAMES campaign.

The strongest DRE values attributed to the above boundary layer NPF, MSOA, DMS and ship emission factors are during the summer season (climax transition (bloom maxima) and declining phase). This result highlights the link between the level of marine biogenic activity and aerosol climate effects. The second strongest individual DRE is attributed to condensing marine organic vapors, yielding MSOA. In our simulations, MSOA contributes significantly to particle growth to diameters of about 100 to 200 nm, which can then interact directly with radiation (Fig. 2). This effect is greatest in the declining phase because we used a temperature-dependent parameterization and sea surface temperatures are warmest during the late summer. The DRE geographic distribution suggests an increasing role for MSOA towards southern latitudes, again due to the temperature-dependent parameterization. Further work is needed to examine the role of MSOA in the more southerly latitudes as we cannot explicitly test this result across the annual cycle with the NAAMES observations.

Among the factors considered, Figure 7 shows that during the time of the May/June phytoplankton bloom, the aerosols produced and grown by the oxidation products of DMS have the third strongest

impact on the DRE, greatest over the regions where the bloom is located. The DRE is -0.10 Wm$^{-2}$
over the region between 40-60 $^{\circ}$N and 20-50 $^{\circ}$W during the bloom maxima and diminishes to -
0.005 Wm$^{-2}$ during the bloom minima. This simulated impact of DMS relies in part on (1)
conditions favoring NPF processes near and above the MBL top, and (2) growth by MSOA as the
nascent DMS-related particles descend in the MBL. DMS (similar to MSOA) also contributes to
the DRE over the continents as these vapors have a lifetime of about a day in our simulations and
can be transported before their oxidation products are available for condensation. Once available
for condensation, DMS products and MSOA contribute to growing particles (of both marine and
continental origin) to sizes that can interact more strongly with radiation (diameters near 100 -200
nm).  Particles arising from DMS grow during transport, and some particles may only reach sizes
large enough to interact with radiation when they are over the continents.

The DRE attributed to the near and above MBL top NPF factor (Fig. 7, top row, ABLNUC) is
strongest in summertime, during the May/June climax transition (bloom maxima) and
August/September declining phase. Summertime is the season of the greatest photochemical
production of particle precursors for NPF. In order to contribute to the DRE, this NPF factor acts
in synergy with the other factors, particularly DMS as a source of particle precursors and MSOA
for particle growth, such that during the May/June climax transition season the DREs for those
factors dominate over the NPF factor (ABLNUC, Fig. 7).

The DREs for ship emissions have a similar geographic distribution as those for DMS. In these
regions, major international ship traffic routes are coincident with regions of higher biogenic
activity, enabling an interaction of biogenic and anthropogenic emissions. Ships enhance oxidant
levels, which promote formation of biogenic aerosol precursors such as sulfuric acid and MSA
that arise from oxidation of DMS. Condensing vapors of marine origin (such as DMS products
and MSOA precursors) can also help to grow particles arising from ship emissions to sizes large
enough to interact directly with radiation. As a result, the largest DRE attributed to ship emissions
is during the phytoplankton bloom maxima. Figure 7 also suggests that ship emissions could
contribute to the DRE over the continents. This effect occurs because ship emissions include
particle precursors, oxidants, and primary particles that are transported and interact with
continental pollution to form and grow particles to sizes that can interact with radiation over the
continents as well as over the oceans. Figure 6 shows that there is a ship-emission-related control
on atmospheric DMS mixing ratios, which increase when the ship-source oxidants are removed.

Figure 8 shows the calculated two-month mean cloud-albedo aerosol indirect effect (AIE)
attributed to each of the same five factors that we considered for the DREs. The AIEs are about an
order-of-magnitude larger than the calculated DRE for each respective factor with the exception
of sea spray. The AIE is strongly controlled by changes to highly reflective MBL clouds, which
are in turn very sensitive to the aerosol number concentrations (diameters larger than about 50 to
70 nm that can act as CCN), which are controlled by the MBL-related factors examined here. On
the other hand, the DRE is relatively more sensitive to aerosol abundance in mid-tropospheric
layers, which are less influenced by the considered processes.

The strongest simulated AIEs for all considered factors are during the May/June climax transition
(Fig. 8). There is a strong synergy among all factors that reach their maxima during May/June
when the effective combination of sources, photochemistry and particle production/growth
processes peak. As well, during summertime, the magnitude of the AIE for all factors is greater in
the more northward regions of the North Atlantic relative to more southerly latitudes. These more
northerly regions are less influenced by continental pollution and have lower CCN concentrations,
coupled with persistent low cloud cover. These conditions make these regions quite sensitive to
the factors controlling MBL aerosol size distributions studied here.

In all seasons, we calculated a stronger AIE related to (1) NPF near and above the MBL top
(ABLNUC, top row, Fig. 8) and (2) MSOA (contributor to particle growth) than to (1) DMS (2)
ship emissions and (3) sea spray emissions. In our simulations, the major source of CCN-sized
particles in the North Atlantic MBL during the summer is particle nucleation near and above the
MBL top with growth by MSOA. Without either of these factors, the number concentration of
CCN-sized particles in the simulations drops dramatically (Fig. 2). Hence, it is unsurprising that
the largest simulated AIEs are due to these two factors during the summertime (climate transition
and declining phase). The stronger AIEs attributed to NPF near and above the MBL top (Fig. 8,
top row, ABLNUC) relative to DMS and ship emissions indicate that near and above MBL NPF
in our simulations is controlled not only by the sulfuric acid from the oxidation of DMS or ship
SO₂, but also arising from other sources, including $SO_2$ transported from continental sources.
However, the maximum North Atlantic regional-mean AIE attributed to ship emissions (-0.62 W
m⁻² for the May/June climax transition) still exceeds the global mean effect of -0.155 W m⁻²
attributed to international shipping calculated by Jin et al. (2018), showing the strong location-
dependence and seasonality of this factor. Ship emissions enhance the oxidation rate of DMS, such
that the largest AIE attributed to ships occurs during the phytoplankton bloom due to increased
particle formation/growth during this season.

In our simulations, sea spray has a lower contribution to aerosol number concentrations, among
the factors considered, and as a result has the smallest AIEs. However, recent studies have pointed
out that there are knowledge gaps related to the sea spray emissions parameterizations (e.g Bian et
al., 2019; Regayrre et al., 2020). Future work is needed to gain confidence in the magnitude of the
AIE attributed to sea spray.

We caution that both the DRE and AIE calculations represent a relative contribution of the
considered factors to climate effects in the North Atlantic. However, further work is needed to gain
confidence in the absolute magnitudes. The activation-style nucleation, which we used as a proxy
for the unknown nucleation mechanisms above the marine boundary layer, contributes uncertainty
to the climate effects of this nucleation. There is much more work that needs to be done regarding
the role of MSOA in this system. Certainly, if MSOA is contributing directly to NPF, it would
increase MSOA's climatic importance. However, we have little knowledge of the MSOA
precursor species, their chemical lifetimes, and their role in NPF, so we did not explore these
dimensions in the study. Like the DRE estimates, the separate AIEs are not linearly additive. Other
aerosol indirect effects related to changes in cloud lifetime and precipitation are the subject of
future work. In summary, these calculated DREs and AIEs suggest that aerosol-climate impacts
for North Atlantic regions are controlled by a combination of strong biogenic and anthropogenic
influences and that the nucleation near and above the MBL top contributes to important radiative
effects.

**4. Conclusions**

In this study, we examined aerosol size distribution and composition measurements from the
NAAMES campaigns. These ship and aircraft campaigns took place over four separate stages of
the annual cycle of marine biogenic activity in the Northwest Atlantic during 2015-2018. We used
the GEOS-Chem-TOMAS model with size-resolved aerosol microphysics to interpret these
NAAMES measurements. Observations in layers of the lower troposphere below 6 km showed
enhancements in the campaign-median number concentration of particles with diameters between
3-10 nm. These enhancements indicated new particle formation, and were most pronounced during
the May/June 2016 climax transition (phytoplankton bloom maxima) in the lowest 2 km of
atmosphere, particularly near and just above the boundary layer top. This lower tropospheric
region near and above the MBL top is a key region for marine NPF. This zone above the MBL
clouds is generally very clean, which favors both NPF and strongly sunlit, which favors the
photochemical oxidative production of particle precursors for NPF. The November 2015 winter
transition (phytoplankton bloom minima) was characterized by the lowest particle number
concentrations. During the summer months, OM, followed by sulfate mass concentrations made
strong contributions the total aerosol loading in the lowest 2 km. However, sea spray dominated
the MBL aerosol mass loading. Peak near-surface sulfate concentrations occurred in May/June
during the phytoplankton bloom, whereas peak near-surface OM concentrations were in
August/September. Campaign-median MBL aerosol size distributions were dominated by Aitken
mode particles (diameters 10-100 nm) during the summertime (May/June climax transition and
August/September declining phase). The larger accumulation mode particles were dominant in the
November winter transition and March/April accumulation phase.

Our simulations suggested that a synergy of key factors contributed to Northwest Atlantic MBL
aerosol size distributions, including (1) new particle formation near and above the MBL top; (2)
growth of the newly formed particles by condensation of marine organic vapors, forming marine
secondary organic aerosol (MSOA), which yields more abundant CCN-sized particles that descend
into the MBL while continuing to grow and being subject to cloud processing (e.g., aqueous-phase
aerosol production, which does not add to particle number); (3) DMS-oxidation products that
contribute to particle formation and growth; (4) ship emissions, which are a source of primary and
secondary particles and also contribute to atmospheric oxidants and (5) sea spray emissions, which
also provide a condensation sink that suppresses particle formation. These findings are in
agreement with previous observational-based studies for the North Atlantic region (e.g., Sanchez
et al., 2018; Zheng et al., 2020)

We calculated the aerosol direct (DRE) and cloud-albedo indirect (AIE) radiative effects over the
North Atlantic attributed to five key factors controlling MBL aerosols. The cooling effects were
about a factor of 10 larger for the AIEs than the respective DREs except for sea spray, which
dominated the DRE. The strong AIE response was attributed to the strong sensitivity of the MBL
cloud reflectivity to the MBL-related factors that we examined. Mid-tropospheric aerosol (altitude
of transport of continental pollution) has a strong impact on the DRE and the factors that we
considered had less impact at these altitudes. The maximum regional-mean (40-60 $^o$N, 20-50 $^o$W)
DRE for our simulations was -1.37 W m$^{-2}$, attributed to sea spray during the March/April
accumulating phase, which is a time of strong synoptic-scale storms in the Northwest Atlantic,
enhancing wind-generated sea spray. This strong DRE attributed to sea spray highlights the
importance of work to better constrain parameterizations for models. The second strongest DRE
was connected to the temperature-dependent source of MSOA, which had a key role in growing
simulated particles to large enough (diameters of 100-200 nm) to strongly reflect incoming solar
radiation. The maximum AIE was -3.37 W m$^{-2}$, for the May/June climax transition phase (peak
phytoplankton bloom). This AIE was related to the role MSOA in growing new particles to CCN
sizes as they descend into the MBL and are subject to further growth in clouds after their formation
near the MBL top. The AIE attributed to the NPF factor was nearly as large (-2.27 W m$^{-2}$) during
May/June. The NPF and MSOA factors act in concert with each other and removal of either of
these factors contributed to shutdown the production of cloud-condensation-nuclei-size particles.
Our study demonstrated acceptable model-measurement agreement for our base simulation, such
that our simulations can be employed to examine the *potential* role and relative importance of the
considered factors in the DRE and AIE. However, we caution that further work is needed to gain
confidence in the absolute magnitudes. In particular, the activation-style nucleation, which we
used as a proxy for the unknown nucleation mechanism above the marine boundary layer, adds
uncertainty to the climate effects of this nucleation

This study highlighted the importance of processes connected to both marine biogenic activity and
anthropogenic activity in controlling aerosol size distributions in the Northwest Atlantic MBL. We
identified key factors, which could be the focus of future work. Particularly, work is needed to
better understand the nature, flux, and chemistry of marine organic vapors that can form MSOA.
As well, work is needed to better understand the contributors to NPF near and above the MBL top.
Further work is also needed to understand the interactions of the considered factors with cloud
processing of aerosols and its relative importance in particle growth. As the Earth's climate
changes and shipping traffic/regulations/routes change, work to understand the source strength of
DMS, MSOA, shipping and sea spray emissions is highly relevant. Such work will bridge the
knowledge gaps related to factors controlling aerosols in the marine MBL and their climate
impacts.

**Code and data availability.** The NAAMES project website is at https://naames.larc.nasa.gov. The
NAAMES airborne and ship datasets used in this paper are publicly available and permanently
archived in the NASA Atmospheric Science Data Center (ASDC;
https://doi.org/10.5067/Suborbital/NAAMES/DATA001) and the SeaWiFS Bio-Optical Archive
and Storage System (SeaBASS; https://doi.org/10.5067/SeaBASS/NAAMES/DATA001). The
ship datasets generated during and analyzed for NAAMES studies are also available in the UCSD
Library Digital Collection repository, https://doi.org/10.6075/J04T6GJ6. The GEOS-Chem model
is freely available for download at https://github.com/geoschem/geos-chem (last access 19 July

2020).


**Supplement link.**

**Author contributions.** BC, RVM and JRP designed the study. BC conducted the GEOS-Chem-
TOMAS simulations, led the related analysis, and wrote the manuscript with contributions from
all coauthors. RHM, ECC, and LDZ contributed the aerosol measurements from aboard the NASA
C130 aircraft. AW, MM and AS contributed the gas-phase measurements from aboard the NASA
C130 aircraft. LMR and GS contributed the aerosol measurements from aboard the R/V Atlantis.
RYWC and HL contributed to the interpretation of model-measurement comparisons. EEM
contributed the CEDS data set. KRB contributed to the off-line radiative calculations, MG
contributed the satellite DMS data set.

**Competing interests.** The authors declare that they have no conflict of interest.

**Acknowledgements**. BC, RVM and RYWC acknowledge research funding provided by the
Ocean Frontier Institute, through an award from the Canada First Research Excellence Fund.
JRP and KRB acknowledge funding support from the Monfort Excellence Fund, and the US
Department of Energy's Atmospheric System Research, an Office of Science, Office of
Biological and Environmental Research program, under grant DE-SC0019000. LMR and GS
acknowledge funding support from NASA grant NNX15AE66G. RHM, ECC, LDZ, LMR, and
GS acknowledge funding support from the NASA NAAMES EVS-2 project. HL acknowledges
funding support from the NASA NAAMES mission and the National Institute of Aerospace's
IRAD program. DMS measurements aboard the NASA C-130 during NAAMES were supported
by the Austrian Federal Ministry for Transport, Innovation and Technology (bmvit) through the
Austrian Space Applications Programme (ASAP) of the Austrian Research Promotion Agency
(FFG). MM's participation in NAAMES 2016 was funded by the Tiroler Wissenschaftsfonds
(grant # UNI-0404/1895). AS's participation in NAAMES 2017 was partly funded by National
Institute of Aerospace (Task No 80LARC18F0031). MG acknowledges funding support from the
Natural Sciences and Engineering Research Council of Canada through the NETCARE project
of the Climate Change and Atmospheric Research Program. Tomas Mikoviny is acknowledged
for technical support; Ionicon Analytik is acknowledged for instrumental support

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

 **Tables and figure captions and tables**

| Simulation | Description |
|---|---|
| **BASE** | Control simulation with GEOS-Chem-TOMAS model (GCT12.1.1) as described in Sect. 2.2 |
| **noABLNUC** | Same as BASE, excluding the surrogate activation-type particle nucleation parameterization above the marine boundary layer to about 2 km altitude, as described in Sect. 2.2 |
| **noMSOA** | Same as BASE, excluding the temperature-dependent marine organic vapors, forming marine secondary organic aerosol (MSOA) |
| **noDMS** | Same as BASE, excluding all emissions of DMS |
| **noSHIPS** | Same as BASE, excluding all ship emissions |
| **noSS** | Same as BASE, excluding all sea spray emissions |


**Table 1**: GEOS-Chem-TOMAS simulation acronyms. Simulations and methodology are
described in detail in Sect. 2.2 and 2.3.

| Simulation | Nov 2015 Bloom Minima | May/June 2016 Bloom Maxima | Aug/Sept 2017 Declining Phase | Mar/Apr 2018 Accumulating | Annual Mean |
|---|---|---|---|---|---|
| **BASE** | 0.20 | 0.33 | 0.04 | 0.28 | 0.21 |
| **noABLNUC** | 0.95 | 0.51 | 0.89 | 0.50 | 0.71 |
| **noMSOA** | 0.76 | 0.31 | 0.84 | 0.59 | 0.63 |
| **noDMS** | 0.44 | 0.27 | 0.43 | 0.06 | 0.30 |
| **noSHIPS** | 0.31 | 0.13 | 0.23 | 0.21 | 0.22 |
| **noSS** | 0.31 | 0.24 | 0.12 | 0.28 | 0.24 |



**Table 2:** Mean fractional error (MFE) between observations and the six GEOS-Chem-TOMAS
simulations described in Sect. 2.2 and Table 1 for the ship-track campaign-median aerosol size
distributions shown in Fig. 2.

**Figure 1:** Cruise and aircraft tracks for the 2015-2018 NAAMES campaigns. Flight altitudes
below 3 km are color-coded in medium blue and above 3 km in red. Ship tracks campaigns are
color-coded for each year as shown by the legend, and as follows: Orange: November 2015
winter transition (bloom minima); Cyan: May/June 2016 climax transition (bloom maxima);
Purple: August/September 2017 declining phase; Green: March/April 2018 accumulation phase.

**Figure 2:** NAAMES cruise-track campaign-median marine boundary layer aerosol size
distributions from marine-influenced SEMS (particle diameters 20-500 nm) observations (black,
with 25[th] to 75[th] percentiles in grey) and for the six GEOS-Chem-TOMAS simulations as
described in Table 1 (color-coded as shown in legend).

**Figure 3:** Vertical profiles of NAAMES campaign-median integrated SMPS observations aboard
aircraft at standard temperature and pressure (STP) for particles with diameters of 10 to 282 nm
(black, with 25[th]-75[th] percentiles in grey) and for the six GEOS-Chem-TOMAS simulations
described in Table 1 (color-coded as shown in legend). All measurement and model output is
binned at 500 m resolution and campaign-median values plotted at the mid-point of each bin
starting at 250 m above the surface. Lines show linear interpolation between these values.
**Figure 4:** Vertical profiles of NAAMES campaign-median total number concentrations for
particles with diameters larger than 3 nm (N3), 10 nm (N10) and between 3 to 10 nm (N3-N10)
from CPC observations aboard aircraft at standard temperature and pressure (STP) (black, with
25[th]-75[th] percentiles in grey) and for the six GEOS-Chem-TOMAS simulations described in Table
1 (color-coded as shown in legend). All measurement and model output is binned at 500 m
resolution and campaign-median values are plotted at the mid-point of each bin starting at 250 m
above the surface. Lines show linear interpolation between these values.

**Figure 5:** Vertical profiles of NAAMES campaign-median aerosol non-refractory sulfate and
organic mass concentrations at standard temperature and pressure (STP) from Aerosol Mass
Spectrometer and refractory black carbon from Single Particle Soot Photometer observations
aboard aircraft (black, with 25[th]-75[th] percentiles in grey) and for the six GEOS-Chem-TOMAS
simulations described in Table 1 (color-coded as shown in legend). Simulated sulfate shown is

non-sea-salt-sulfate. All measurement and model output is binned at 500 m resolution and campaign-median values are plotted at the mid-point of each bin starting at 250 m above the surface. Lines show linear interpolation between these values.

**Figure 6:** Vertical profiles of NAAMES cruise-track campaign-median observed dimethyl sulfide (DMS) mixing ratios (black, 25[th]-75[th] percentiles in grey) from aboard aircraft and for the six GEOS-Chem-TOMAS simulations described in Table 1 (color-coded as shown in legend). Simulations BASE, noABLNUC, noMSOA and noSS are nearly coincident. All measurement and model output is binned at 500 m resolution and campaign-median values plotted at the mid-point of each bin starting at 250 m above the surface. Lines show linear interpolation between these values. Note the horizontal scale change between panels.

**Figure 7:** GEOS-Chem-TOMAS-simulated two-monthly-mean aerosol direct radiative effect (DRE) attributed to five key factors. Top row: Above boundary layer particle nucleation (ABLNUC); Second row: Particle growth by marine secondary organic aerosol (MSOA); Third row: Particle formation/growth due to DMS-oxidation products (DMS); Fourth row: Shipping emissions contribution to particles (SHIPS); Bottom row: Sea spray (SS). DREs are in columns for the following time periods, March/April 2018 (Accumulating Phase), May/June 2016 (Climax Transition, Bloom Maxima), August/September 2017 (Declining Phase), and October/November 2015 (Winter Transition, Bloom Minima). DREs for ABLNUC, MSOA, DMS, SHIPS, and SS are calculated using the differences in the top-of-the-atmosphere solar flux between simulation BASE and respective sensitivity simulations (noABLNUC, noMSOA, noDMS, noSHIPS, noSS). Values shown are area-weighted-mean DREs over the region bounded by 40-60 °N, 20-50 °W.

**Figure 8:** GEOS-Chem-TOMAS-simulated two-monthly-mean aerosol cloud-albedo indirect radiative effect (AIE) attributed to five key factors. Top row: Above boundary layer particle nucleation (ABLNUC); Second row: Particle growth by marine secondary organic aerosol (MSOA); Third row: Particle formation/growth due to DMS-oxidation products (DMS); Fourth row: Shipping emissions contribution to particles (SHIPS); Bottom row: Sea spray (SS). AIEs are in columns for the following time periods, March/April 2018 (Accumulating Phase), May/June 2016 (Climax Transition, Bloom Maxima), August/September 2017 (Declining Phase), and

October/November 2015 (Winter Transition, Bloom Minima). AIEs for ABLNUC, MSOA, DMS,
SHIPS, and SS are calculated using the differences in the top-of-the-atmosphere solar flux between
simulation BASE and respective sensitivity simulations (noABLNUC, noMSOA, noDMS,
noSHIPS, noSS). Values shown are area-weighted-mean AIEs over the region bounded by 40-60
$^{o}$N, 20-50 $^{o}$W.
























