# Peer review of "Factors controlling marine aerosol size distributions and"

_Atmospheric Chemistry and Physics, 2020_

## Referee Comment (RC1) · Anonymous Referee #1 · 23 Sep 2020

The manuscript provides an overview of particle number and composition measurements over remote marine locations and aims at explaining sources of these particles by deploying the GEOS-Chem-TOMAS model. While it is a really unique set of measurements that is very difficult to obtain over remote locations, especially extending over various seasons, I feel that the model results are often over-interpreted. Particularly, for the cases where model simulations do not quite agree with the measurements, but important claims are made from these anyway. The manuscript has a potential, owing to the measurements, to bring valuable insights and provide new knowledge, however, several key issues must be resolved before publication.

[Figure]

My main concern is that the paper seems to be biased to discussing and evaluating sources that were selected based on the literature, but not on the current measurements.

To name a few: new particle formation near the surface was not accounted for, nor primary marine sources were seriously considered. E.g. sea salt and sea spray, including organic matter, can be as small as 10 nm in diameter (de Leeuw et al., 2011), however, their effects to DRE and AIE were not evaluated, nor their contribution to vertical distributions analysed. Near surface or MBL NPF was completely omitted, while authors admit that MSOA NPF, which supposedly happens near the surface, can potentially have an impact (lines 275-277). The question is then why they were not included or evaluated? Without the proper evaluation, their role cannot be dismissed and conclusions stating that only cloud base or MBL top NPF are important are based on wrong assumptions.

Moreover, the conclusion on the dominant near MBL top NPF effect was based on very uncertain measurements near the ground. As the reliability of near ground measurements poses some questions, e.g. how the aircraft data were extrapolated to the ground level as, I assume, the airplane did not go down to 0m altitude? And, to my understanding from the methods section, the ship measurements were for particles >20nm in diameter? Therefore, Figure 4 showing N3 and N10 down to 0m as well as lines 406-408 are misleading as they do not represent the real N measurements.

Moreover, in high biological activity May/June period, the N3-N10 maximum extends over the broad amplitude range ($\sim$2km wide), as opposed to only the top of the MBL as stated by authors (408-412), therefore, the question is whether the drop in concentration that occurs at the very surface is due to measurement uncertainty? In which case, the NPF might have occurred over the whole boundary layer and not only at the MBL top, but was not detected due to these measurement limitations?

While, if indeed real, such strong gradient in number concentration would imply a constant source that occurs over larger geographical areas as these new particles are not mixed down into boundary layer within normal 15-20 min mixing, which would diminish the gradient if the source is not constant. The existence of such strong constant and wide source does not seem likely.

The statement on the number concentration maxima observed at 1 km (lines 30-32) is also not robust. The total number concentration, in Figure 3, shows 2 peaks, one at ~1km and other just below 2 km, with the latter being even stronger, so why 1 km maxima is highlighted and how the second maxima is explained? Is that the measurement uncertainty or just noise rather than the real maxima? Similarly, lines 396-397 point to one maximum in figure 3, which is quite subjective as there are many 'maximas' in that figure, but no explanation is provided.

Therefore, the statement 'that NPF near/above the MBL has a strong control on the development of the total particle number maxima near 1 km altitude...' (lines 464-467) is not exactly based on the measurements. Moreover, neither Base nor noABLNUC simulations agree with the measurements, actually, for this season, all N3-N10 simulations are very far from the measurements, so the claims on the processes influencing the number concentrations in the high biological activity season are not substantiated by the measurements or data in this paper. Similarly, the following statement on MBL-top NPF influencing the concentrations of near surface particles (lines 476-748) is not based on the findings as, without model simulations agreeing with the measurements, these are only speculations. There are other features in the simulations that were not properly discussed or explained, like high DMS contribution in winter (Figure 2 and Figure 7, lines 562-565). Provide quantitative (%) estimates when talking about differences between Base and noDMS. It is not so trivial to judge by just looking at the graphs. Also, high DMS effects (as well as MSOA) over continents are not discussed (figure 7).

Finally, the ship emission control does not look so modest to me (lines 593-594 and Figure 2), so, please provide a quantitative estimate of the difference. Also, it seems,

that noSHIPS agrees better with the measurements than the BASE in figure 2? Discus that in more details.

Also, why the ship emission effect (Figure 7 and 8) is the highest for the high biological period, discuss the observed seasonality in detail. How do you explain ship effects over continents? High AIE effect (lines 693-695) might as well point to over-prediction of ship influence rather than location-dependence? This would explain better agreement with measurements in Figure 2 when Ship emissions were not included?

Specific comments: Figure 4: noABLNUC is not visible in some panels, e.g. bottom and middle panels in the right column, please adjust colours or patterns.

Ship measurements do not cover particles smaller than 20nm, how can Figure 4 show the concentrations down to the ground level? Surely the aircraft could not have measured at these low altitudes. Were the measurements extrapolated then? How reliable are these extrapolations? Similarly, for Figure 5, how do composition measurements extend to the ground? Provide details on the extrapolation in the method section.

How the total number in figure 3 over August/September can be reconciled with Figure 4 data for the same phase for N10. N20 from fig 3 shows higher concentrations near the ground with reducing trend towards boundary layer top, which would resemble what is expected for a winter boundary layer with sea salt contributions, but N10 shows the opposite trend. Explain this in more details.

de Leeuw, G., Andreas, E. L., Anguelova, M. D., Fairall, C. W., Lewis, E. R., O'Dowd, C., Schulz, M., and Schwartz, S. E.: Production flux of sea spray aerosol, Rev Geophys, 49, RG2001, doi:10.1029/2010rg000349, 2011.

———————————————

---

## Referee Comment (RC2) · Anonymous Referee #2 · 8 Oct 2020

The paper by Croft et al. provides an assessment of size distributions and it's controls over the Northwest Atlantic Ocean. Climate effects of selected processes are included as forcings, both direct and 1st indirect effect. The work is very valuable: marine aerosol background, especially aerosol size distribution, is not well constrained in climate models. While the analysis in the manuscript is rather standard, the paper is generally well written and results are presented in a clear way. Below are detailed comments on selected issues with the research itself. Some issues listed below should become more clear in the revised manuscript.

The study applies activation-type nucleation (l. 265) with linear function of sulfuric acid

[Figure]

concentrations. The empirical activation nucleation coefficient (A=2*10ˆ6 s-1) is re-trieved in continental environments, and is known to include high variability even over land. As discussed in the same paragraph (l. 275), the role of marine organic com-pounds in nucleation remains unclear, which also has implications in using continental empirical coefficient A in marine environment. This might be very essential for the study, especially since ABLNUC seems to produce a significant AIE (Fig. 8).

Also, is the activation-type nucleation really active only between MBL-top and 2 km altitude (l. 265)? Why not through all levels in MBL? Although large-scale models have typically limited a separate nucleation mechanisms, such as activation-type nucleation to the BL, it seems even more unphysical to limit a mechanism to only a few (not well defined?) regions of the model system. Perhaps another mechanism/parameterization could better account for different regimes in the surface/BL/BL-top/2-km/free tropo-sphere system.

The choice and reasoning behind MSOA emission parameters remains non-conclusive. Five sets of two parameters for MSOA source are simulated, and size dis-tributions are used in constraining the best possible parameter set (e.g. Fig. S1). I do not see that the information compiled in Figs. S1-S4, or even the statistics in Table S1, would convince the chosen source parameters as the best plausible set. Considering the amount of additional assumptions for MSOA, e.g. volatility and chemical procesing (l. 321) as well as dismissing the effect on NPF, the uncertainty in simulated MSOA and conclusions on the respective aerosol-cloud effects remains non-satisfactory.

One key factor when analyzing the role of nucleation or early growth is to constrain the background aerosol, in this case sea spray aerosol, to have e.g. realistic sink described for NPF. According to l. 298, sea salt emissions are simulated according to Jaeglé et al. (2011). Jaeglé et al. (2011) compares several formulations of sea spray emission. Assuming this refers to the flux as a function of SST, Jaeglé et al. (2011) shows that this decreases global emissions from 5200 to 4600 Tg/yr when compared to Gong (2003), with clear reduction over study area. However, e.g. Regayre et al. (2020) used

a massive amount of observations was to constrain a global model with Gong (2003) sea spray flux, and the process indicated that simulated sea spray flux was even a factor of 3 too low. Perhaps this result is not applicable to the study region used here, but some discussion on potential biases could be discussed in model description or in conclusions.

The simulations are using a very coarse horizontal resolution, 4x5 degrees. Even while the observation/model comparison is using hourly output, there is no discussion on potential biases or sampling issues with large grid-scale. Is there any horizontal interpolation performed when sampling the model data against flight/ship data?

References:

Regayre, L. A., Schmale, J., Johnson, J. S., Tatzelt, C., Baccarini, A., Henning, S., Yoshioka, M., Stratmann, F., Gysel-Beer, M., Grosvenor, D. P., and Carslaw, K. S.: The value of remote marine aerosol measurements for constraining radiative forcing uncertainty, Atmos. Chem. Phys., 20, 10063–10072, https://doi.org/10.5194/acp-20-10063-2020, 2020.

---

## Author Response (AR1)

**Author note:** This document contains the following 4 sections (each with page numbering starting with page 1):
1) all responses to both referees as also posted on the ACPD site for acp-2020-811 (pages 1-17)
2) a marked-up version of our revised manuscript (pages 1-66)
3) the revised manuscript without markups (pages 1-66)
4) a marked-up version of the revised supplement (pages 1-22)

**Section 1:**

*We thank the two referees for their comments, which have led to considerable improvements in our manuscript.*

**Response to Anonymous Referee #1**

*The authors thank Anonymous Referee #1 for this detailed review, which has resulted in revisions that have strengthened the manuscript. Below are our point-by-point responses, showing how we have addressed each of the referee's concerns. All referee comments are in black bold text, preceded by RC. Author comments are in blue italics and preceded by AC. All line numbers quoted below refer to the manuscript version without markups.*

**RC: The manuscript provides an overview of particle number and composition measurements over remote marine locations and aims at explaining sources of these particles by deploying the GEOS-Chem-TOMAS model. While it is a really unique set of measurements that is very difficult to obtain over remote locations, especially extending over various seasons, I feel that the model results are often over-interpreted. Particularly, for the cases where model simulations do not quite agree with the measurements, but important claims are made from these anyway. The manuscript has a potential, owing to the measurements, to bring valuable insights and provide new knowledge, however, several key issues must be resolved before publication.**

*AC: In response to the reviewer's comments, we have carefully revised the manuscript to avoid over-interpretation of the model results. We agree that these are a really unique set of measurements over various seasons that can provide valuable insights. Details of the revisions conducted in response to this review are provided below.*

**RC1: My main concern is that the paper seems to be biased to discussing and evaluating sources that were selected based on the literature, but not on the current measurements. To name a few: new particle formation near the surface was not accounted for, nor primary marine sources were seriously considered. E.g. sea salt and sea spray, including organic matter, can be as small as 10 nm in diameter (de Leeuw et al., 2011), however, their effects to DRE and AIE were not evaluated, nor their contribution to vertical distributions analysed.**

*AC1: Thank you for this constructive comment. We agree that the manuscript would be improved by discussion and evaluation of the role of both new particle formation (NPF) near the surface, and primary marine sources (sea salt and sea spray).*

*To address these concerns, we added a section on sea spray to the main text (lines 719-740, Sect. 3.6), and we added a new simulation with no sea spray emissions (noSS, Table 1). As well, in the supplement, we now include a simulation with 3-fold scaling up of sea spray emissions. The revised Sect. 3.6 discusses the revised Figs. 2-5, which now include the noSS simulation. These revised figures enable evaluation of sea-spray-related contributions to both the vertical profiles and the MBL ship-track size distributions. Figures 7 and 8 in the radiative effects section (new Sect. 3.7) were also revised to present an evaluation of sea spray contributions to the DRE and AIE.*

*The abstract was revised to include the following as related to sea spray during the May/June phytoplankton bloom maxima, "and (5) primary sea spray emissions (AIE: +0.04 W m$^{-2}$, DRE: -0.79 W m$^{-2}$)" (lines 43-44).*

*The conclusion was also revised to state "The maximum regional-mean (40-60 $^o$N, 20-50 $^o$W) DRE for our simulations was -1.37 W m$^{-2}$, attributed to sea spray during the March/April accumulating phase, which is a time of strong synoptic-scale storms in the Northwest Atlantic, enhancing wind-generated sea spray" (lines 906-909).*

*Although sea spray contributes to particles as small as 10 nm in diameter in our simulations, we caution that, "A coupled parameterization for primary organic aerosol from sea spray was not available for our aerosol size-resolved GEOS-Chem-TOMAS simulations, such that some sea spray organics could be misrepresented as sea salt" (lines 288-291).*

*To address the role of new particle formation near the surface, we revised the text to clarify that our simulations do account for NFP near the surface. The revised text states that, "All simulations include particle nucleation in the boundary layer that is parameterized with the ternary (H$_2$SO$_4$-NH$_3$-H$_2$O) scheme of Napari et al. (2002), which was scaled by 10$^{-5}$ to better match continental boundary-layer measurements (Westervelt, 2013) (lines 315-317).*

*As well, we added Supplementary Figs. S5-S8 and Supplementary Table S3, which show the effect of extending our surrogate nucleation parameterization through the entire MBL. The following related text was also added to the main manuscript, "Extending the surrogate activation-style parameterization to the surface (Supplementary Figs. S5-S8 and Supplementary Table 3), leads to overprediction of the number of particles with diameters less than 50 nm in the MBL and yields higher MFEs (ranging from 0.20 to 0.56) than for simulation BASE, although the errors were not as large as those for noABLNUC. For the vertical profiles, this extra NPF extended into the MBL yields overprediction of N3, N10, and N3-N10 below 1 km in all seasons. Aerosol surface area and volume (in the SMPS particle-diameter size range of 10 nm - 282 nm) were over predicted during the August/September declining phase, when the simulated temperature-dependent MSOA source was strongest, growing these extra new particles to larger sizes. These challenges highlight the relevance of ongoing research to better understand NPF in the marine environment" (lines 544-553).*

**RC2: Near surface or MBL NPF was completely omitted, while authors admit that MSOA NPF, which supposedly happens near the surface, can potentially have an impact (lines 275-277). The question is then why they were not included or**

**evaluated? Without the proper evaluation, their role cannot be dismissed and conclusions**
**stating that only cloud base or MBL top NPF are important are based on wrong**
**assumptions.**
*AC2: We agree that the manuscript would benefit from improved clarity regarding MBL NPF in*
*our simulations. To address this concern, we have revised the text to more clearly indicate that*
*all simulations do include MBL NPF using a standard ternary scheme (quoted in AC1, lines 315-*
*317).*
*To evaluate the impact of including the surrogate nucleation scheme throughout the MBL, we*
*added Supplementary Figs. S5-S8 and Supplementary Table S3 and related discussion in the*
*main text (lines 544-553, quoted in AC1).*
*As well, we revised the text to include the following evaluation of the ternary NPF scheme in the*
*MBL, "Without the surrogate NPF scheme employed near and above the MBL top, the ternary*
*NPF scheme in the MBL in simulation noABLNUC fails to simulate sufficient particle number,*
*although vertical-profile campaign-median ammonium concentrations below 4 km altitude had*
*acceptable agreement with observations (MFE ranges from 0.12 to 0.48, not shown). Figure 4*
*shows about a one-order-of-magnitude underprediction of N3 below about 2 km for noABLNUC.*
*NoABLNUC has an unacceptable seasonal-mean model-measurement agreement across the*
*measurement set (MFE ranges from 0.66 to 0.78, Supplementary Table S2)" (lines 503-510).*
**RC3: Moreover, the conclusion on the dominant near MBL top NPF effect was based on**
**very uncertain measurements near the ground. As the reliability of near ground**
**measurements poses some questions, e.g. how the aircraft data were extrapolated to the**
**ground level as, I assume, the airplane did not go down to 0m altitude?**
*AC3: To address this concern, we revised the manuscript to provide greater clarity about the near-*
*ground measurements. During NAAMES, the lowest aircraft flight level altitude was around 150-*
*200 m GPS altitude. We revised the text as follows to offer support for the reliability of the near-*
*ground measurement.*
*We revised the methods section to explicitly indicate the approach that was used for binning the*
*measurements into various altitude ranges, "For consideration of vertical profiles, we binned the*
*measurement and simulation values using a 500 m height resolution, starting from the surface to*
*500 m as the first bin. Campaign-median values are calculated within each bin and plotted at the*
*mid-point of the bin, starting at 250 m. During NAAMES, the lowest aircraft flight level altitude*
*was around 150-200 m GPS altitude. We use a plane-flight diagnostic in the model to sample the*
*simulation interpolated between grid-cell centers to the aircraft flight track position during the*
*times when measurement data was available for each respective instrument. We find consistent*
*results with bin resolutions of 250, 500 and 1000 m, giving support for our selected binning*
*resolution. The vertical profiles show measurements and model output along the aircraft flight*
*tracks only and do not include any measurements or model output for the ship track" (lines 376-*
*386).*

**RC4: And, to my understanding from the methods section, the ship measurements were for**
**particles >20nm in diameter? Therefore, Figure 4 showing N3 and N10 down to 0m as well**
**as**
**lines 406-408 are misleading as they do not represent the real N measurements.**

*AC4: Thank you for noting the need for greater clarity in regard to the particle size ranges for*
*Fig. 2 (ship measurements) relative to Fig. 4 (aircraft measurements). To address this concern,*
*we revised lines 414-415 to state "Figure 2 shows the campaign-median marine-influenced*
*aerosol size distributions from SEMS (particle diameters 20-500 nm) for the four R/V Atlantis*
*cruises".*

*We also revised the text in the methods section to clarify the sizes ranges for the two CPCs and*
*to better explain the terminology used, "As well, we give attention to measurements of total*
*particle number concentration from the Condensation Particle Counters (CPCs) with differing*
*nominal lower detection diameters: 3 nm for the CPC 3025 (yielding N3 measurements) and 10*
*nm for the CPC 3772 (TSI Inc., St. Paul, MN) (yielding N10 measurements) aboard the C130*
*aircraft" (lines 227-231).*

*As well, lines 439-441 related to Fig. 4 were revised to clarify that, "Figure 4 shows the vertical-*
*profile campaign-median total particle number concentrations from CPCs, for aerosols with*
*diameters larger than 3 nm (N3), larger than 10 nm (N10), and the difference between the two*
*(N3-N10)".*

*To improve clarity, we also revised the Fig. 4 caption (and all other vertical profile captions) to*
*state the range of the altitude bins, "All measurement and model output is binned at 500 m*
*resolution and campaign-median values are plotted at the mid-point of each bin starting at 250*
*m above the surface". As well, captions for all vertical-profile figures were revised to state that*
*the presented measurements were taken aboard aircraft.*

*As quoted in AC3, the revised text also indicates that these vertical profiles do not include ship-*
*board measurements (lines 384-386).*

**RC5: Moreover, in high biological activity May/June period, the N3-N10 maximum extends**
**over the broad amplitude range (~2km wide), as opposed to only the top of the MBL**
**as stated by authors (408-412), therefore, the question is whether the drop in concentration**
**that occurs at the very surface is due to measurement uncertainty?**

*AC5: We agree that rewording related to the altitude range would improve the clarity and*
*accuracy of the discussion. To address this concern, we revised the text to indicate May/June*
*N3-N10 maximum extends over a broad range, "For the May/June 2016 climax transition*
*(phytoplankton bloom maximum), there are enhancements in observed number concentration*
*(N3, N10 and N3-N10) below about 2 km in the free troposphere, indicating NPF at these*
*altitudes (Fig, 4)" (lines 441-443).*

*To support that the drop in N3-N10 concentration towards the surface was not due to*
*measurement uncertainty, we revised the text to provide additional details (as quoted in AC3 and*
*AC4) about the method used for preparing the vertical profiles.*
**RC6: In which case, the NPF might have occurred over the whole boundary layer and not**
**only at the MBL top, but was not detected due to these measurement limitations?**
*AC6: We agree that NPF can occur over the whole boundary layer. For clarity, the revised text*
*acknowledges that, "NPF does occur in the MBL. However, those levels above the MBL clouds*
*favor oxidative chemistry that yields particle precursors, particularly from the wide-spread and*
*persistent DMS sources in the marine environment (Kazil et al., 2011)" (lines 537-539).*
**RC7: While, if indeed real, such strong gradient in number concentration would imply a**
**constant source that occurs over larger geographical areas as these new particles are not**
**mixed down into boundary layer within normal 15-20 min mixing, which would diminish**
**the gradient if the source is not constant. The existence of such strong constant and**
**wide source does not seem likely.**
*AC7: Thank you for this discussion. We have revised the main text as noted in AC3-AC6 to*
*provide support and discussion related to the number gradient, while being careful to indicate*
*that NPF can occur at all levels. The revised text indicates that our study supports DMS as a*
*relatively constant source of particle precursors, which extends over a large geographic area as*
*quoted in AC6 (lines 537-539).*
*As well, the revised text states that, "The lower free tropospheric region near and above the*
*MBL top is an important region for marine NPF. These altitudes above the MBL clouds are*
*generally very clean, which favors NPF, and strongly sunlit, which favors the photochemical*
*oxidative production of particle precursors for NPF" (lines 444-448) and with regard to our*
*simulations that "Although our simulations do include NPF within the MBL, simulated NPF*
*occurs more strongly near and above the MBL top and the resultant particles grow by*
*condensation of available vapors and cloud processing while descending into the MBL. This role*
*for NPF is in agreement with previous studies including those of Clarke et al. (2013), Quinn et*
*al. (2017), and Williamson et al. (2019)" (lines 531-535).*
**RC8: The statement on the number concentration maxima observed at 1 km (lines 30-32) is**
**also not robust. The total number concentration, in Figure 3, shows 2 peaks, one at**
**_1km and other just below 2 km, with the latter being even stronger, so why 1 km maxima**
**is highlighted and how the second maxima is explained? Is that the measurement**
**uncertainty or just noise rather than the real maxima?**
*AC8: We agree that these statements brought too much focus to the 1 km level. To address this*
*concern, we have revised the abstract text to state, "Observations from the NAAMES campaigns*
*show enhancements in the campaign-median number of aerosols with diameters larger than 3*
*nm in the lower troposphere (below 6 km), most pronounced during the phytoplankton bloom*
*maxima (May/June) below 2 km in the free troposphere" (lines 30-33).*

*The text regarding Fig.3 was also revised to state, "These profiles exhibit several particle*
*number maxima in the lower free troposphere below 6 km, including below 2 km during the*
*May/June climax transition period" (lines 428-430).*
*The revised text also provides additional details about the binning method used for the vertical*
*profiles, which offers support that the maxima in the presented campaign-median profiles are not*
*noise.*
**RC9: Similarly, lines 396-397 point to one maximum in figure 3, which is quite subjective**
**as there are many 'maximas' in that figure, but no explanation is provided.**
*AC9: We agree that the discussion would be improved by indicating that there can be several*
*maxima in the lower free troposphere. The text is revised as quoted in AC8 (lines 30-33 and lines*
*428-430). The text was also revised to explain that, "As shown in Fig. 3, aerosol surface area*
*and volume are less at altitudes below about 3 km, relative to altitudes above 3 km. This lower*
*particle surface area at these altitudes favors NPF over growth of pre-existing particles as*
*available vapors condense in these relatively cleaner atmospheric layers (Kazil et al., 2011).*
*Transport of aerosols (in part associated with continental emissions) contributes to particles in*
*all seasons. Fast et al. (2016) characterized summertime North Atlantic transport layers in the*
*free troposphere associated with synoptic-scale lifting" (lines 430-436).*
*The revised text also discusses evidence for levels where NPF contributes relatively more*
*strongly to the number maxima and states, "For the May/June 2016 climax transition*
*(phytoplankton bloom maximum), there are enhancements in observed number concentration*
*(N3, N10 and N3-N10) below about 2 km in the free troposphere, indicating NPF at these*
*altitudes (Fig. 4)" (lines 441-443).*
**RC10: Therefore, the statement 'that NPF near/above the MBL has a strong control on the**
**development of the total particle number maxima near 1 km altitude...' (lines 464-467)**
**is not exactly based on the measurements.**
*AC10: We agree that the original text was overly focused on the 1 km altitude and that the total*
*particle number maxima could have a variety of contributors.  To address this concern, we have*
*revised the text as quoted in AC9 to acknowledge that there are a variety of contributors to the*
*particle maxima, and that measurements suggest the strongest role for NPF in the lowest 2 km in*
*May/June phytoplankton bloom period.*
**RC11: Moreover, neither Base nor noABLNUC simulations agree with the measurements,**
**actually, for this season, all N3-N10 simulations are very far from the measurements, so the**
**claims on the processes influencing the number concentrations in the high biological**
**activity season are not substantiated by the measurements or data in this paper.**
*AC11: In response to this comment, we have revised the text to acknowledge that the N3-N10 is*
*very challenging to the model, "The simulated N3-N10 (Fig. 4) illustrates that representation of*
*NPF is a challenge for models, because there are difficulties capturing the magnitude and*
*altitudes of the N3-N10 maxima. These discrepancies reflect key knowledge gaps related to the*

*species that can form new particles in the marine environment (e.g., Veres et al. 2020). As well,*
*the coefficient that we used for the surrogate activation-style nucleation parameterization was*
*derived for a continental environment. The empirical ('A') value used by the parameterization*
*appears to yield excessive NPF for the NAAMES marine environment. Activation-style*
*nucleation was added in our simulations as a proxy for missing nucleation when the*
*condensation sink is low, and conditions favor high oxidation rates. We acknowledge that this*
*approach will miss variability in the timing and rates because it is a surrogate and not exactly*
*the correct mechanism. As well in the summertime, the simulations underpredict N3-N10*
*concentrations above 2km, suggesting the need for future work to better understand the NPF*
*processes at these levels, where the binary scheme of Vehkamaki et al. (2002) does not generate*
*sufficient NPF" (lines 516-528).*
*However, we consider that simulation BASE has acceptable performance such that it can be*
*used for interpreting these measurements. For more complete evaluation of simulation BASE*
*relative to noABLNUC, we extended our calculation of MFEs to all panels presented in the main*
*text. These MFEs are summarized in the new Supplementary Table S2 and indicate that BASE*
*offers an overall seasonal-mean acceptable simulation (mean MFE ranges from 0.43-0.50) and*
*represents an improvement over noABLNUC (mean MFE ranges from 0.66-0.78). The text was*
*revised to state, "For each season the mean MFE across the parameters considered in Figs. 2 to*
*5 (BASE versus measurements, Supplementary Table S2) is satisfactory (MFE ranges 0.43 to*
*0.50)" (lines 492-494). Also, the revised text states," NoABLNUC has an unacceptable seasonal-*
*mean model-measurement agreement across the measurement set (MFE ranges from 0.66 to*
*0.78) (Supplementary Table S2)" (lines 508-510).*
*We consider that our study can help to direct future research efforts by identifying key*
*challenges in marine aerosol simulation. At the same time, simulation BASE shows satisfactory*
*model-measurement agreement, which supports conclusions made by comparing the simulation*
*set with measurements, such as the following revised statement, "Without the surrogate NPF*
*scheme employed near and above the MBL top, the ternary NPF scheme in the MBL in*
*simulation noABLNUC fails to simulate sufficient particle number, although vertical-profile*
*campaign-median ammonium concentrations below 4 km altitude had acceptable agreement with*
*observations (MFE ranges from 0.12 to 0.48, not shown). Figure 4 shows about a one-order-of-*
*magnitude underprediction of N3 below about 2 km for noABLNUC" (lines 503-508).*
**AC12: Similarly, the following statement on MBLtop NPF influencing the concentrations**
**of near surface particles (lines 476-748) is not based on the findings as, without model**
**simulations agreeing with the measurements, these are only speculations.**
*AC12: To address this concern, we added additional quantitative analysis as noted in AC11 and*
*presented in Supplementary Table S2. This analysis indicates that across the measurement set,*
*simulation BASE provides acceptable agreement with the measurements (MFE ranges from 0.43*
*to 0.50 for seasonal mean across the measurement set), unlike noABLNUC (MFE ranges from*
*0.66 to 0.78 for seasonal mean across the measurement set). This analysis is supportive of the*
*role of NPF near and above the MBL top as a key contributor to the MBL size distributions.*

*We also revised the text to clarify how we defined acceptable model-measurement agreement.*
*Lines 373-374 state, "We adopt the convention of Boylan and Russell (2006) to consider a MFE*
*of 0.5 or less as acceptable."*

**AC13: There are other features in the simulations that were not**
**properly discussed or explained, like high DMS contribution in winter (Figure 2 and**
**Figure 7, lines 562-565). Provide quantitative (%) estimates when talking about differences**
**between Base and noDMS. It is not so trivial to judge by just looking at the**
**graphs. Also, high DMS effects (as well as MSOA) over continents are not discussed**
**(figure 7).**

*AC13: Thank you for pointing out the need for greater clarity in the discussion related to DMS.*
*To address these concerns, we revised the original lines 562-565 to quantify the difference*
*between BASE and noDMS and to clarify that there is not a high contribution of DMS in the*
*winter. The revised text states, "Figure 2 shows that DMS also has a control on the simulated*
*MBL aerosol size distributions (BASE versus noDMS) for the four seasons of the NAAMES*
*campaigns. The total simulated number of particles attributed to DMS is lowest during the*
*phytoplankton bloom minima (winter, November 2015) and greater in other seasons. For*
*example, for particle diameters at 40 nm, the DMS-related contribution to the size distribution*
*(Fig. 2) is about 200-300 $cm^{-3}$ in all seasons, except less than 50 $cm^{-3}$ during the bloom minima"*
*(lines 654-659)".*

*We revised the caption of Fig. 6 to caution, "Note the horizontal scale change between panels".*

*As well, we revised the text related to DMS and Fig. 7 to clarify/quantify that "The DRE is -0.10*
*$Wm^{-2}$ over the region between 40-60 $^o$N and 20-50 $^o$W during the bloom maxima and diminishes*
*to -0.005 $Wm^{-2}$ during the bloom minima" (lines 773-775).*

*We also revised the text to include discussion about the DMS-related radiative effects over the*
*continents. The revised text states, "DMS (similar to MSOA) also contributes to the DRE over the*
*continents as these vapors have a lifetime of about a day in our simulations and can be transported*
*before their oxidation products are available for condensation. Once available for condensation,*
*DMS products and MSOA contribute to growing particles (of both marine and continental origin)*
*to sizes that can interact more strongly with radiation (diameters near 100 -200 nm. Particles*
*arising from DMS grow during transport, and some particles may only reach sizes large enough*
*to interact with radiation when they are over the continents" (lines 777-783).*

**RC14: Finally, the ship emission control does not look so modest to me (lines 593-594 and**
**Figure 2), so, please provide a quantitative estimate of the difference. Also, it seems,**
**that noSHIPS agrees better with the measurements than the BASE in figure 2? Discuss**
**that in more details.**

*AC14: We agree with this comment and have removed the word 'modest' from the discussion*
*and added a quantitative statement about the differences between noSHIPS and BASE. The*
*revised text states that "Our simulations suggest that ship emissions are also a control on the*
*NAAMES-region MBL marine-influenced aerosol size distributions (Fig. 2, noSHIPS versus*

*BASE). For example, for the simulated summertime MBL size distribution at particle diameters*
*at 40 nm, about 100-200 cm$^{-3}$ are attributed to ship emissions (Fig. 2)" (lines 689-693).*
*We also added the following discussion about the model-measurement agreement for noSHIPS*
*versus BASE, "Table 2 shows that during the phytoplankton bloom and March/April*
*accumulating phase, the noSHIPS simulation agrees more closely with the measurements than*
*the BASE simulation, although both are within acceptable agreement (MFE < 0.5). These*
*simulation challenges highlight the importance of future work to better understand the role of*
*oxidants from ship emissions on particle production in the marine environment and to*
*understand the size distribution of primary marine emissions" (lines 693-698).*
**RC15: Also, why the ship emission effect (Figure 7 and 8) is the highest for the high**
**biological**
**period, discuss the observed seasonality in detail.**
*AC15: We revised the text to comment more clearly on the seasonality of the ship effects. The*
*text states, "Ships enhance oxidant levels, which promote formation of biogenic aerosol*
*precursors such as sulfuric acid and MSA that arise from oxidation of DMS. Condensing vapors*
*of marine origin (such as DMS products and MSOA precursors) can also help to grow particles*
*arising from ship emissions to sizes large enough to interact directly with radiation. As a result,*
*the largest DRE attributed to ship emissions is during the phytoplankton bloom maxima" (lines*
*795-800).*
*The text related to the AIE was also revised to include the following discussion, "Ship emissions*
*enhance the oxidation rate of DMS, such that the largest AIE attributed to ships occurs during*
*the phytoplankton bloom due to increased particle formation/growth during this season" (lines*
*839-841).*
**RC16: How do you explain ship effects over continents?**
*AC16: We added the following discussion related to effects over the continents. "Figure 7 also*
*suggests that ship emissions could contribute to the DRE over the continents. This effect occurs*
*because ship emissions include both particle precursors, oxidants, and primary particles that are*
*transported and interact with continental pollution to form and grow particles to sizes that can*
*interact with radiation over the continents as well as over the oceans" (lines 800-804).*
**RC17: High AIE effect (lines 693-695) might as well point to over-prediction of ship**
**influence rather than location-dependence? This would explain better agreement with**
**measurements in Figure 2 when Ship emissions were not included?**
*AC17: We revised the text to indicate that noSHIPS agrees more closely with the measurements*
*than BASE in some seasons as related to Fig 2, "Table 2 shows that during the phytoplankton*
*bloom and March/April accumulating phase, the noSHIPS simulation agrees more closely with*
*the measurements than the BASE simulation, although both are within acceptable agreement*
*(MFE < 0.5) (lines 693-695)".*

*As a result, Fig. 2 does not provide conclusive support that the ship emissions are over predicted. As noted in AC15, this stronger regional AIE compared to the global mean could be related to the interaction of the ship emissions with the products of the phytoplankton bloom maxima in this region. However, as acknowledged in AC14, there were simulation challenges related to ship emissions, which require future study.*

**RC18: Specific comments: Figure 4: noABLNUC is not visible in some panels, e.g. bottom and middle panels in the right column, please adjust colours or patterns.**

*AC18: In response to this comment, we have revised Fig. 4 to more clearly identify simulation noABLNUC by adding square symbols to the plot for ABLNUC. Other figures were also revised for consistency. Thank you for noting this problem with visibility.*

**RC19:  Ship measurements do not cover particles smaller than 20nm, how can Figure 4 show**
**the concentrations down to the ground level? Surely the aircraft could not have measured at these low altitudes. Were the measurements extrapolated then? How reliable are these extrapolations?**

*AC19: To address this concern we revised the caption of Fig. 4 to clarify that these profiles do show aircraft measurements, not ship measurements. We also added details about the aircraft flight altitudes and the method used for binning of the measurements by altitude as quoted in AC3. The reliability of the measurements is supported by this additional discussion regarding our methodology.*

**AC20: Similarly, for Figure 5, how do composition measurements**
**extend to the ground? Provide details on the extrapolation in the method section**

*AC20: Thank you for noting the need for greater clarity in the discussion about the methodology for production of the vertical profile figures. These details are added to the revised text as noted in AC3-AC5. Also, the caption for Fig. 5 was revised to clarify the binning resolution used.*

**RC21: How the total number in figure 3 over August/September can be reconciled with Figure**
**4 data for the same phase for N10. N20 from fig 3 shows higher concentrations near the ground with reducing trend towards boundary layer top, which would resemble what is expected for a winter boundary layer with sea salt contributions, but N10 shows the opposite trend. Explain this in more details.**

*AC21: The SMPS (Fig. 3) and CPC (Fig. 4) were operated during similar times, but there were some differences in the times of data availability from the SMPS and CPC that could contribute to the differences between these 2 figures. However, for all model-measurement comparisons, the model was only sampled at times when a measurement from the respective instrument was available. This is now clarified in the method with the following revised text, "We use a plane-flight diagnostic in the model to sample the simulation interpolated between grid-cell centers to*

*the aircraft-flight-track position during the times when measurement data was available for each*
*respective instrument" (lines 379-382).*
**Response to Anonymous Referee #2**
*The authors thank Anonymous Referee #2 for this constructive review. The following point-by-*
*point responses indicate how we have addressed each of the referee's concerns. All referee*
*comments are in black bold text, preceded by RC. Author comments are in blue italics and*
*preceded by AC. All line numbers quoted below refer to the manuscript version without markups.*
**RC: The paper by Croft et al. provides an assessment of size distributions and it's controls**
**over the Northwest Atlantic Ocean. Climate effects of selected processes are**
**included as forcings, both direct and 1st indirect effect. The work is very valuable:**
**marine aerosol background, especially aerosol size distribution, is not well constrained**
**in climate models. While the analysis in the manuscript is rather standard, the paper**
**is generally well written and results are presented in a clear way. Below are detailed**
**comments on selected issues with the research itself. Some issues listed below should**
**become more clear in the revised manuscript.**
*AC: Thank you for these comments. In response to this review, we have conducted revisions that*
*improve the clarity of the presentation, strengthening the manuscript. Details are provided in*
*our point-by-point responses below.*
**RC1: The study applies activation-type nucleation (l. 265) with linear function of sulfuric**
**acid**
**concentrations. The empirical activation nucleation coefficient (A=2\*10ˆ6 s-1) is retrieved**
**in continental environments, and is known to include high variability even over**
**land. As discussed in the same paragraph (l. 275), the role of marine organic compounds**
**in nucleation remains unclear, which also has implications in using continental**
**empirical coefficient A in marine environment. This might be very essential for the**
**study, especially since ABLNUC seems to produce a significant AIE (Fig. 8).**
*AC1: Thank you for this comment about the empirical nucleation coefficient. We revised the*
*manuscript to include the following discussion, "As well, the coefficient that we used for the*
*surrogate activation-style nucleation parameterization was derived for a continental*
*environment. The empirical ('A') value used by the parameterization appears to yield excessive*
*NPF for the NAAMES marine environment. Activation-style nucleation was added in our*
*simulations as a proxy for missing nucleation when the condensation sink is low, and conditions*
*favor high oxidation rates. We acknowledge that this approach will miss variability in the timing*
*and rates because it is a surrogate and not exactly the correct mechanism." (lines 519-525).*
*We also added the following clarification statement related to the AIE, "We caution that both the*
*DRE and AIE calculations represent a relative contribution of the considered factors to climate*
*effects in the North Atlantic. However, further work is needed to gain confidence in the absolute*
*magnitudes. The activation-style nucleation, which we used as a proxy for the unknown nucleation*

*mechanisms above the marine boundary layer, adds uncertainty to the climate effects of this*
*nucleation. Certainly, if MSOA is contributing directly to NPF, it would increase MSOA's climatic*
*importance. However, we have little knowledge of the MSOA precursor species, their chemical*
*lifetimes, and their role in NPF, so we did not explore these dimensions in the study." (lines 849-*
*857).*
*Despite the noted uncertainty in activation nucleation coefficient, the revised text now more clearly*
*indicates that simulation BASE (with the activation-type nucleation) yields acceptable model-*
*measurement agreement, "For each season the mean model-measurement MFE across the*
*parameters considered in Figs. 2 to 5 (BASE versus measurements, Table S2) is satisfactory (MFE*
*ranges 0.43 to 0.50)" (lines 492-494). The text also now more clearly indicates that the model*
*performance is unacceptable for noABLNUC, "Without the surrogate activation-style NPF*
*scheme employed near and above the MBL top, the ternary NPF scheme in the MBL in simulation*
*noABLNUC fails to simulate sufficient particle number, although vertical-profile campaign-*
*median ammonium concentrations below 4 km altitude had acceptable agreement with*
*observations (MFE ranges from 0.12 to 0.48, not shown). Figure 4 shows about a one-order-of-*
*magnitude underprediction of N3 below about 2 km for noABLNUC. NoABLNUC has an*
*unacceptable seasonal-mean model-measurement agreement across the measurement set (MFE*
*ranges 0.68 to 0.78, Supplementary Table S2)" (lines 503-510).*
*The text was also revised to clarify that we consider this surrogate activation-style scheme to be*
*a place-holder until related knowledge gaps are resolved, "The extra nucleation in the lower*
*troposphere with the activation-type parameterization represents particle precursors that could*
*have the same source as sulfuric acid. This approach may not capture the timing and magnitude*
*of the variability in NPF correctly because the vapors participating in this nucleation are likely*
*not just sulfuric acid. Future work is needed to better understand the nature of the nucleating*
*species in the lower troposphere over the oceans" (lines 335-340).*
**RC2: Also, is the activation-type nucleation really active only between MBL-top and 2 km**
**altitude (l. 265)? Why not through all levels in MBL?**
*AC2: Yes, in the original manuscript, we only had this nucleation scheme between the MBL-top*
*and 2 km altitude because it was clear that there was an enhancement of N3-10 that we were not*
*capturing in the model. This scheme was added as a proxy for the unknown mechanism. To*
*address the reviewer's question about having the activation mechanism throughout the MBL, we*
*added Supplementary Figs. S5-S8 and Supplementary Table S3. These figures and table show the*
*impact of extending the activation-type nucleation throughout the MBL. This approach leads to*
*over prediction of the particle number, surface area and volume (in the SMPS particle diameter*
*size range of 10-282 nm) in the MBL.*
*The revised text clarifies that, "Extending the surrogate activation-style parameterization to the*
*surface (Supplementary Figs. S5-S8 and Supplementary Table 3), leads to over prediction of the*
*number of particles with diameters less than 50 nm in the MBL and yields higher MFEs (ranging*
*from 0.20 to 0.56) than for simulation BASE, although the errors were not as large as those for*
*noABLNUC. For the vertical profiles, this extra NPF extended into the MBL yields over prediction*
*of N3, N10, and N3-N10. Aerosol surface area and volume (in the SMPS particle-diameter size*

*range of 10 nm - 282 nm) were also over predicted during the August/September declining phase,*
*when the simulated temperature-dependent MSOA source was strongest, growing these extra new*
*particles to larger sizes. These challenges highlight the relevance of ongoing research to better*
*understand NPF in the marine environment" (lines 544-553).*
*We also revised the methods to clarify that all simulations do include nucleation in the MBL,*
*"All simulations include particle nucleation in the boundary layer that is parameterized with the*
*ternary ($H_2SO_4$-$NH_3$-$H_2O$) scheme of Napari et al. (2002), which was scaled by $10^{-5}$ to better*
*match continental boundary-layer measurements (Westervelt et al., 2013)" (lines 315-317).*
**RC3: Although large-scale models have typically limited a separate nucleation**
**mechanisms, such as activation-type nucleation to the BL, it seems even more unphysical to**
**limit a mechanism to only a few (not well defined?) regions of the model system. Perhaps**
**another mechanism/parameterization could better account for different regimes in the**
**surface/BL/BL-top/2-km/free troposphere system.**
*AC3: In this study we tested the impact of added particle nucleation in the lower troposphere*
*over the oceans, between the MBL top and 2 km. This was motivated by the occurrence of the*
*largest measurement N3-N10 concentrations below 2 km (Fig. 4, phytoplankton bloom), and this*
*enhancement being not captured by the model without the added nucleation. The revised text*
*points out that "For the May/June 2016 climax transition (phytoplankton bloom maximum),*
*there are enhancements in observed number concentration (N3, N10 and N3-N10) below about 2*
*km in the free troposphere, indicating NPF at these altitudes (Fig, 4). The MBL top ranged from*
*about 0.5 to 2 km for the NAAMES cruises (Behrenfeld et al., 2019). The lower free tropospheric*
*region near and above the MBL top is an important region for marine NPF. These altitudes*
*above the MBL clouds are generally very clean, which favors NPF, and strongly sunlit, which*
*favors the photochemical oxidative production of particle precursors for NPF" (lines 441-448).*
*However, the model-measurement agreement deteriorates when the parameterization was*
*extended through the entire MBL, as indicated by the revised text that is quoted in AC2*
*We also revised the text to indicate the need for future work to better understand and parameterize*
*NPF at various altitudes in the marine environment, "As well in the summertime, the simulations*
*underpredict N3-N10 concentrations above 2 km, suggesting the need for future work to better*
*understand the NPF processes at these levels, where the binary scheme of Vehkamaki et al. (2002)*
*does not generate sufficient NPF" (lines 525-528).*
**RC4: The choice and reasoning behind MSOA emission parameters remains**
**nonconclusive.**
**Five sets of two parameters for MSOA source are simulated, and size distributions**
**are used in constraining the best possible parameter set (e.g. Fig. S1). I do**
**not see that the information compiled in Figs. S1-S4, or even the statistics in Table S1,**
**would convince the chosen source parameters as the best plausible set.**

AC4: To address this concern, we revised Supplementary Figures S1-S4 and Supplementary
Tables S1 and S2. Figures S1-S4 and Table S1 were revised to include a simulation with no
MSOA, and a simulation with the chosen parameterization scaled up by a factor of 10.
These revised figures and table provide clarification regarding the choice and reasoning behind
the MSOA emission parameterization. This additional information supports that the chosen
parameters are the best for the various emission schemes that we tried, have acceptable MFEs.
and are physically plausible.

The related discussion to the main text was revised to state, "For the NAAMES MBL size
distributions, the annual-mean model-measurement MFEs are acceptable (ranging from 0.23 –
0.38, lowest for BASE) for all temperature-dependent parameterizations that we tested, except for
the factor-of-ten scaling up of the BASE MSOA parameterization (simulation 10x(70T-350),
Supplementary Table S1, MFE of 0.75) and with the MSOA parameterization removed (simulation
noMSOA, Supplementary Table S1, MFE of 0.63). While this source flux is reasonably constrained
for our simulations, future work is needed to better understand and parameterize this source"
(lines 588-594).

As well, the revised text states, "The vertical profiles are also sensitive to the MSOA
parameterization (Supplementary Figs. S2-4). Between noMSOA and the various MSOA
parameterizations that we tested, concentrations vary by up to a factor of about 2 for aerosol
number (N3, N10, and N3-N10), SMPS-size-range (diameters 10 nm --282 nm) number, surface
area, volume and also OM. Simulation noMSOA has relatively greater error in the mean across
the entire measurement set for each season (MFE ranges from 0.53-0.68) relative to BASE (MFE
ranges from 0.42-0.50) (Supplementary Table S2)" (lines 596-601).

We also added the following discussion about the plausibility of our MSOA parameterization,
"Although the chosen MSOA parameterization reasonably represents the observations, major
knowledge gaps remain regarding MSOA precursor species and their chemical lifetimes. While
the nature of MSOA precursors is not well-understood, recent measurements suggest that these
precursors could include a variety of chemical compounds. For example, measurements from the
Arctic indicate that the organics in marine aerosols were not typical biogenic SOA but had a long-
hydrocarbon chain implying a fatty acid type precursor (Willis et al., 2017). In other marine
regions, isoprene (Ciuraru et al., 2015) and carboxylic acids (Chiu et al., 2017) may also be
important. Given the limitations of current knowledge and the indications for a variety of MSOA
precursors, the improved MFEs for BASE relative to noMSOA provide support for the employed
MSOA parameterization" (lines 603-612).

**RC5: Considering the amount of additional assumptions for MSOA, e.g. volatility and
chemical processing (l. 321) as well as dismissing the effect on NPF, the uncertainty in
simulated MSOA
and conclusions on the respective aerosol-cloud effects remains non-satisfactory.**

AC5: We agree that there is much more work that needs to be done regarding the role of MSOA
in this system. Certainly, if MSOA is contributing directly to NPF, it would increase MSOA's
climactic importance. However, we have little knowledge of the MSOA precursor species, their
chemical lifetimes, and their role in NPF, so we did not explore these dimensions in the study.

*We consider that our study demonstrates acceptable model-measurement agreement for*
*simulation BASE, such that our simulations can be employed to examine the potential role of*
*MSOA on AIE. We added metrics as outlined in AC4 to support our use of simulation BASE,*
*which includes MSOA. We also revised the text to highlight the need for future work as quoted in*
*AC4 (lines 603-612).*

*As well, we added cautionary words about uncertainty in the magnitude of the AIE as quoted in*
*AC1 (lines 849-857).*

*We also added the following cautionary statement in the conclusion, "Our study demonstrated*
*acceptable model-measurement agreement for our base simulation, such that our simulations*
*can be employed to examine the potential role and relative importance of the considered factors*
*in the DRE and AIE. However, we caution that, further work is needed to gain confidence in the*
*absolute magnitudes. In particular, the activation-style nucleation, which we used as a proxy for*
*the unknown nucleation mechanism above the marine boundary layer, adds uncertainty to the*
*climate effects of this nucleation" (lines 919-924).*

**RC6: One key factor when analyzing the role of nucleation or early growth is to constrain**
**the**
**background aerosol, in this case sea spray aerosol, to have e.g. realistic sink described**
**for NPF. According to l. 298, sea salt emissions are simulated according to Jaeglé et**
**al. (2011). Jaeglé et al. (2011) compares several formulations of sea spray emission.**
**Assuming this refers to the flux as a function of SST, Jaeglé et al. (2011) shows that**
**this decreases global emissions from 5200 to 4600 Tg/yr when compared to Gong**
**(2003), with clear reduction over study area. However, e.g. Regayre et al. (2020) used**
**a massive amount of observations was to constrain a global model with Gong (2003)**
**sea spray flux, and the process indicated that simulated sea spray flux was even a**
**factor of 3 too low. Perhaps this result is not applicable to the study region used here,**
**but some discussion on potential biases could be discussed in model description or in**
**conclusions.**

*AC6: Thank you for this discussion about the role of sea spray aerosol. We have added an*
*analysis of this species' role. We revised the manuscript and supplement to include additional*
*simulations with no sea spray emissions (noSS) and with 3-fold scaling up of the sea spray*
*emissions (3xSS). This additional analysis related to sea spray is included in the new Sect. 3.6*
*(lines 719-740).*

*We also added the following comment to the introduction to more clearly acknowledge potential*
*biases, "Recent studies have highlighted knowledge gaps related to sea spray emissions,*
*particularly as related to the submicron sizes (e.g. Bian et al., 2019; Regayre et al, 2020).*
*Measurement and modeling studies are needed to better understand and simulate the size-*
*resolved contribution of sea spray to the Northwest Atlantic MBL" (lines 87-90).*

*As well, we added the following text to the model description for clarification about the*
*performance of our simulations, "Sea salt emissions follow Jaeglé et al. (2011). This*
*temperature-dependent parameterization decreases global emissions relative to the Gong (2003)*

*parameterization. A coupled parameterization for primary organic aerosol from sea spray was*
*not available for our aerosol size-resolved GEOS-Chem-TOMAS simulations, such that some sea*
*spray organics could be misrepresented as sea salt, since all sea spray in our simulations is*
*considered sea salt" (lines 287-291).*
*We added Supplementary Fig. S10, which shows the simulated sea spray mass concentrations for*
*simulation BASE.  The revised text now more clearly indicates that our simulations have a*
*reasonable representation of the sea spray emissions, while commenting on potential biases as*
*follows, "The simulated campaign-median MBL sea spray mass concentrations are within the*
*measurement range of 3-8 µg m$^{-3}$ d$^{-1}$ found by Saliba et al. (2019) (Supplementary Fig. S10),*
*despite the considerable uncertainties related to size-resolved sea spray emissions (e.g. Bian et*
*al., 2019; Regayre et al. (2020)). Regayre et al. (2020) found that global sea spray emissions*
*could be under predicted by a factor of 3 by the Gong (2003) parameterization. We conducted a*
*simulation with factor-of-3 scaling of the sea spray emissions (Supplementary Figs. S11-S14,*
*Supplementary Table S4) and found a decrease in MBL number concentrations, rather than an*
*increase. This reduction occurred because the enhanced condensation sink from the additional*
*sea spray emissions suppressed NPF. Our simulations use the Gong (2003) parameterization*
*with a sea-surface-temperature-based scaling as described by Jaeglé et al. (2011), so are not*
*directly comparable to the Regayre et al. (2020) findings. Nonetheless, these findings highlight*
*the importance of ongoing work to improve size-resolved sea spray emissions parameterizations*
*in models" (lines 727-739).*
*We added the following discussion related to the DRE attributed to sea spray, "Figure 7*
*indicates that the strongest calculated DRE is attributed to sea spray, which dominates the*
*aerosol mass loading in the MBL. The sea spray DRE has a maximum during the 2018*
*March/April accumulating phase, which is a time of frequent synoptic scale storms with strong*
*winds. Stormy conditions prevented the R/V Atlantis from travelling north of 45 ºN during this*
*final NAAMES campaign" (lines 752-756).*
*We also added the following discussion related to the AIE attributed to sea spray, "In our*
*simulations, sea spray has a lower contribution to aerosol number concentrations, among the*
*factors considered, and as a result has the smallest AIEs. However, recent studies have pointed*
*out that there are knowledge gaps related to the sea spray emissions parameterizations (e.g Bian*
*et al., 2019; Regayre et al., 2020). Future work is needed to gain confidence in the magnitude of*
*the AIE attributed to sea spray" (lines 843-847).*
*The revised conclusion also notes that, "This strong DRE attributed to sea spray highlights the*
*importance of work to better constrain parameterizations for models" (lines 909-910).*
**RC7: The simulations are using a very coarse horizontal resolution, 4x5 degrees. Even**
**while the observation/model comparison is using hourly output, there is no discussion**
**on potential biases or sampling issues with large grid-scale.**
*AC7: Thank you for pointing out this omission. To address this concern, we revised the methods*
*section to include the following discussion about the potential for biases and sampling issues as*
*related to model grid-scale, "To manage computational expense, the simulations are necessarily*

*at a coarse resolution, which can bias model-measurement comparisons. However, these biases will be lower for remote marine regions such as the NAAMES study region than over land regions, which generally have greater spatial inhomogeneity. Representativeness errors were also reduced by limiting our model-measurement comparisons to campaign-median values" (lines 402-406).*

**RC8: Is there any horizontal interpolation performed when sampling the model data against flight/ship data?**

*AC8: Thank you for pointing out the need for clarification about our methodology for sampling the model. We added the following text to clarify the methods that we used for model sampling along the flight tracks, "
[revised manuscript text omitted]
 mean fractional error (MFE, 0.5 or less, Boylan and Russell, 2006) between NAAMES campaign-median measurements and simulations (described in Sect. 2). The simulated MBL aerosol size distributions for a set of sources fluxes are shown in Supplementary Fig. S1. Supplementary Table S1 shows the MFEs for this set of source fluxes. Among this set of source fluxes, we found the lowest annual mean MFE for the source flux of 70T-350 (T in $^{o}$C and flux in $\mu$g m$^{-2}$ d$^{-1}$). We caution that this tuning was specific for the NAAMES region and for a certain GEOS-Chem-TOMAS model configuration. As a result, this source flux may not perform as well in other models, other GEOS-Chem versions and other regions.

All of the temperature dependent parameterizations shown (Supplementary Fig.S1) had acceptable annual mean MFEs for the MBL size distributions, with the exception of 1) the simulation with the factor-of-ten scaling up of the flux that was used in BASE (10x(70T-350)) and 2) the simulation without condensable marine organic vapors (noMSOA). We consider that the order of magnitude of the flux was reasonably constrained for the purposes of this study under the various emission schemes that we tried. However, our findings suggest that further work is needed to better constrain the flux of marine organic vapors.

The selected parameterization also yielded agreement within the 25$^{th}$ to 75$^{th}$ percentiles for the campaign-median vertical profiles in the lowest 1 km for total aerosol number (N3, N10 and N3-N10) and integrated SMPS number, and near-surface OM concentrations (Supplementary Figs. S2-S4). Supplementary Fig. S2 shows slight overprediction outside of these percentiles for the integrated SMPS surface area and volume below 2 km. For the vertical profiles, the mean MFEs across the measurement set were acceptable for the BASE simulation with the 70T-350 source flux and unacceptable for noMSOA (Supplementary Table 2).

**Section S2: Mean fractional errors** The MFEs for the for all panels of Figs. 2 through 6 in the main text are shown in Supplementary Table S2. For vertical profiles, the MFEs are calculated using a summation (Eq. 1) over the altitude bins.

**Section S3: Role of new particle formation** A sensitivity simulation with the surrogate nucleation parameterization extended to the surface layer (BASE+BLNUC) increased simulated particle number in the MBL relative to BASE, worsening agreement with measurements (Supplementary Figs. S5-S8 and Supplementary Table S3).

**Section S4: Role of ship emissions** We found enhancements in benzene relative to other tracers, such as acetone, which have anthropogenic sources but not associated with ship emissions (Supplementary Fig. S9). These findings are supportive of the study region being influenced by ship emissions.

**Section S5: Role of sea spray**. Simulated campaign-median sea spray mass concentrations (Supplementary Fig. S10) were within the 3-8 in $\mu g\ m^{-3}$ range with a maximum for NAAMES in March/April 2018, which is in agreement with measurements reported by Saliba et al. (2019). Sensitivity studies with no sea spray and sea spray scaled up by a factor of 3 were conducted. In terms of simulated particle number, the factor-of-3 scaling up of sea spray made a stronger contribution as a condensation sink, suppressing the total particle number in the MBL (Supplementary Figs. S11-S14 and Supplementary Table S4).

**Section S1: Role of MSOA**

[Figure]

**Figure S1:** NAAMES cruise-track campaign-median marine boundary layer aerosol size
distributions from marine-influenced SEMS observations (black, with 25[th] to 75[th] percentiles in
grey) and for seven GEOS-Chem-TOMAS simulations with different assumptions for the
temperature dependence of the flux of condensable organic vapors (color-coded as shown in
legend, flux in μg m$^{-2}$ d$^{-1}$ and T in ºC).

| Marine organic vapor source | Nov 2015 Bloom Minima | May/June 2016 Bloom Maxima | Aug/Sept 2017 Declining Phase | Mar/Apr 2018 Accumulating | Annual Mean |
|---|---|---|---|---|---|
| **70T-350 (BASE)** | 0.20 | 0.33 | 0.04 | 0.28 | 0.23 |
| **50T-300** | 0.34 | 0.22 | 0.22 | 0.21 | 0.25 |
| **50T-500** | 0.55 | 0.20 | 0.56 | 0.23 | 0.38 |
| **100T-300** | 0.11 | 0.54 | 0.26 | 0.33 | 0.31 |
| **100T-500** | 0.13 | 0.30 | 0.19 | 0.31 | 0.27 |
| **noMSOA** | 0.76 | 0.31 | 0.84 | 0.59 | 0.63 |
| **10x(70T-350)** | 0.87 | 0.80 | 0.73 | 0.60 | 0.75 |

**Table S1:** Mean fractional error (MFE) between observations and seven GEOS-Chem-TOMAS simulations for the ship-track campaign-median aerosol size distributions shown in Supplementary Fig. S1 (T in $^{\circ}$C and source flux in $\mu$g m$^{-2}$ d$^{-1}$).

[Figure]

**Figure S2:** Vertical profiles of NAAMES campaign-median integrated SMPS observations at
standard temperature and pressure (STP) for particles with diameters of 10 to 282 nm (black, with
25[th]-75[th] percentiles in grey) and at STP for seven GEOS-Chem-TOMAS simulations with
different assumptions for the temperature dependence of the flux of condensable marine organic
vapors (color-coded as shown in legend, flux in $\mu g\ m^{-2}\ d^{-1}$ and T in $^{o}C$). All measurement and
model output are binned at 500 m resolution and campaign-median values plotted at the mid-point
of each bin starting at 250 m above the surface. Lines show linear interpolation between these
values.

[Figure]

**Figure S3:** Vertical profiles of NAAMES campaign-median total number concentrations for
particles with diameters larger than 3 nm (N3), 10 nm (N10) and between 3 to 10 nm (N3-N10)
from CPC observations at standard temperature and pressure (STP) (black, with 25th-75th
percentiles in grey) and at STP for seven GEOS-Chem-TOMAS simulations with different
assumptions for the temperature dependence of the flux of condensable marine organic vapors
(color-coded as shown in legend, flux in μg m$^{-2}$ d$^{-1}$ and T in ºC). All measurement and model
output are binned at 500 m resolution and campaign-median values plotted at the mid-point of
each bin starting at 250 m above the surface. Lines show linear interpolation between these values.

[Figure]

**Figure S4:** Vertical profiles of NAAMES campaign-median aerosol non-refractory sulfate and organic mass concentrations from Aerosol Mass Spectrometer and refractory black carbon from

SP2 observations at standard temperature and pressure (STP) (black, with $25^{th}$-$75^{th}$ percentiles in grey) and at STP for seven GEOS-Chem-TOMAS simulations with different assumptions for the temperature dependence of the flux of marine condensable organic vapors (color-coded as shown in legend, flux in μg m$^{-2}$ d$^{-1}$ and T in ºC). Simulated sulfate shown is non-sea-salt-sulfate. All measurement and model output are binned at 500 m resolution and campaign-median values plotted at the mid-point of each bin starting at 250 m above the surface. Lines show linear interpolation between these values.

**Section S2: Mean fractional errors**

| MFEs for figure panels | BASE | noABLNUC | noMSOA | noDMS | noSHIP | noSS |
|---|---|---|---|---|---|---|
| **2015** | | | | | | |
| MBL size dist. Fig 2 | 0.20 | 0.95 | 0.76 | 0.44 | 0.31 | 0.31 |
| Number Fig 3 | 0.31 | 0.39 | 0.40 | 0.26 | 0.25 | 0.30 |
| Surface area Fig 3 | 0.43 | 0.68 | 0.73 | 0.43 | 0.44 | 0.43 |
| Volume Fig 3 | 1.22 | 1.04 | 0.96 | 1.12 | 1.21 | 1.22 |
| N3 Fig 4 | 0.51 | 0.75 | 1.02 | 0.43 | 0.44 | 0.53 |
| N10 Fig 4 | 0.24 | 0.43 | 0.23 | 0.20 | 0.24 | 0.25 |
| N3-N10 Fig 4 | 0.88 | 1.04 | 1.34 | 0.90 | 0.92 | 0.87 |
| Sulfate Fig 5 | 0.39 | 0.26 | 0.25 | 0.27 | 0.34 | 0.39 |
| Organic mass Fig 5 | 0.52 | 0.51 | 0.73 | 0.52 | 0.52 | 0.53 |
| Black carbon Fig 5 | 0.44 | 0.45 | 0.45 | 0.44 | 0.44 | 0.44 |
| DMS Fig 6 | 0.12 | 0.12 | 0.12 | 0.67 | 0.40 | 0.12 |
| Average of 2015 | 0.42 | 0.68 | 0.67 | 0.50 | 0.46 | 0.45 |
| **2016** | | | | | | |
| MBL size dist. Fig 2 | 0.33 | 0.51 | 0.31 | 0.27 | 0.13 | 0.24 |
| Number Fig 3 | 0.37 | 0.60 | 0.38 | 0.43 | 0.34 | 0.37 |
| Surface area Fig 3 | 1.04 | 1.10 | 1.16 | 1.09 | 1.06 | 1.04 |
| Volume Fig 3 | 0.50 | 0.46 | 0.39 | 0.44 | 0.49 | 0.50 |
| N3 Fig 4 | 0.35 | 0.74 | 0.43 | 0.62 | 0.36 | 0.37 |
| N10 Fig 4 | 0.31 | 0.61 | 0.31 | 0.47 | 0.34 | 0.30 |
| N3-N10 Fig 4 | 1.08 | 1.48 | 1.10 | 1.40 | 1.11 | 1.10 |
| Sulfate Fig 5 | 0.52 | 0.52 | 0.52 | 0.16 | 0.56 | 0.52 |
| Organic mass Fig 5 | 0.60 | 0.61 | 0.84 | 0.59 | 0.60 | 0.60 |
| Black carbon Fig 5 | 0.78 | 0.78 | 0.78 | 0.78 | 0.78 | 0.78 |
| DMS Fig 6 | 0.26 | 0.26 | 0.26 | 0.67 | 0.31 | 0.26 |
| Average of 2016 | 0.50 | 0.66 | 0.53 | 0.55 | 0.46 | 0.49 |
| **2017** | | | | | | |
| MBL size dist. Fig 2 | 0.04 | 0.89 | 0.84 | 0.43 | 0.23 | 0.12 |
| Number Fig 3 | 0.60 | 0.91 | 0.50 | 0.73 | 0.60 | 0.61 |
| Surface area Fig 3 | 0.88 | 1.12 | 1.18 | 1.02 | 0.91 | 0.87 |
| Volume Fig 3 | 0.78 | 0.61 | 0.56 | 0.67 | 0.77 | 0.79 |
| N3 Fig 4 | 0.43 | 0.86 | 0.49 | 0.69 | 0.47 | 0.45 |
| N10 Fig 4 | 0.58 | 0.97 | 0.49 | 0.81 | 0.59 | 0.56 |
| N3-N10 Fig 4 | 1.09 | 1.47 | 1.01 | 1.40 | 1.11 | 1.04 |
| Sulfate Fig 5 | 0.17 | 0.18 | 0.17 | 0.27 | 0.19 | 0.17 |
| Organic mass Fig 5 | 0.65 | 0.65 | 1.03 | 0.63 | 0.66 | 0.66 |
| Black carbon Fig 5 | 0.47 | 0.48 | 0.48 | 0.47 | 0.47 | 0.47 |
| DMS Fig 6 | 0.18 | 0.18 | 0.20 | 0.67 | 0.20 | 0.18 |
| Average of 2017 | 0.43 | 0.78 | 0.68 | 0.65 | 0.49 | 0.45 |
| **Average all years** | 0.45 | 0.71 | 0.63 | 0.57 | 0.47 | 0.46 |

**Table S2:** Mean fractional error (MFE) between the six simulations described in Table 1 and the
measurements for the panels of Figs. 2 through 6. Results for Fig. 2 are weighted to include MFEs
for first four moments of the MBL aerosol size distributions. All MFEs are calculated for altitude
below 6 km, except below 2 km for DMS due to the decrease over orders of magnitude above 2
km. The MFEs are calculated following Eq. 1 with a summation over the altitude bins that are
defined in Sect.2.

**Section S3: Role of new particle formation**

[Figure]

**Figure S5:** NAAMES cruise-track campaign-median marine boundary layer aerosol size
distributions from marine-influenced SEMS observations (black, with 25[th] to 75[th] percentiles in
grey) and for three GEOS-Chem-TOMAS simulations with different assumptions for surrogate
above boundary layer nucleation. noABLNUC: surrogate activation nucleation scheme above the
boundary removed; BASE+BLNUC: surrogate activation nucleation scheme extended from 2 km
to the surface layer; BASE as described in Table 1 and Section 2, including surrogate activation
nucleation scheme from above the boundary layer to 2 km.

|  | Nov 2015 Bloom Minima | May/June 2016 Bloom Maxima | Aug/Sept 2017 Declining Phase | Mar/Apr 2018 Accumulating | Annual Mean |
|---|---|---|---|---|---|
| **BASE** | 0.20 | 0.33 | 0.04 | 0.28 | 0.21 |
| **noABLNUC** | 0.95 | 0.54 | 0.89 | 0.50 | 0.72 |
| **BASE+BLNUC** | 0.36 | 0.56 | 0.20 | 0.48 | 0.40 |

**Table S3:** Mean fractional error between observations and three GEOS-Chem-TOMAS

simulations for the ship-track campaign-median aerosol size distributions shown in Supplementary

Fig. S5.

[Figure]

**Figure S6:** Vertical profiles of NAAMES campaign-median integrated SMPS observations at standard temperature and pressure (STP) for particles with diameters of 10 to 282 nm (black, with $25^{th}$-$75^{th}$ percentiles in grey) and at STP for three GEOS-Chem-TOMAS simulations with different assumptions for above boundary layer nucleation. noABLNUC: surrogate activation nucleation scheme above the boundary removed; BASE+BLNUC: surrogate activation nucleation scheme extended from 2 km to the surface layer; BASE as described in Table 1 and Section 2, including surrogate activation nucleation scheme from above the boundary layer to 2 km. All measurement and model output are binned at 500 m resolution and campaign-median values plotted at the mid-point of each bin starting at 250 m above the surface. Lines show linear interpolation between these values.

[Figure]

**Figure S7:** Vertical profiles of NAAMES campaign-median total number concentrations for
particles with diameters larger than 3 nm (N3), 10 nm (N10) and between 3 to 10 nm (N3-N10)
from CPC observations at standard temperature and pressure (STP) (black, with 25th-75th
percentiles in grey) and at STP for three GEOS-Chem-TOMAS simulations with different
assumptions for above boundary layer nucleation. noABLNUC: surrogate activation nucleation
scheme above the boundary removed; BASE+BLNUC: surrogate activation nucleation scheme
extended from 2 km to the surface layer; BASE as described in Table 1 and Section 2, including
surrogate activation nucleation scheme from above the boundary layer to 2 km. All measurement
and model output are binned at 500 m resolution and campaign-median values plotted at the mid-
point of each bin starting at 250 m above the surface. Lines show linear interpolation between
these values.

[Figure]

**Figure S8:** Vertical profiles of NAAMES campaign-median aerosol non-refractory sulfate and organic mass concentrations from Aerosol Mass Spectrometer and refractory black carbon from SP2 observations at standard temperature and pressure (STP) (black, with $25^{th}$-$75^{th}$ percentiles in grey) and at STP for three GEOS-Chem-TOMAS simulations with different assumptions for above boundary layer nucleation. noABLNUC: surrogate activation nucleation scheme above the boundary removed; BASE+BLNUC: surrogate activation nucleation scheme extended from 2 km to the surface layer; BASE as described in Table 1 and Section 2, including surrogate activation nucleation scheme from above the boundary layer to 2 km. All measurement and model output are binned at 500 m resolution and campaign-median values plotted at the mid-point of each bin starting at 250 m above the surface. Lines show linear interpolation between these values.

**Section S4: Role of ship emissions**

[Figure]

**Figure S9:** Vertical profiles of NAAMES campaign-median benzene (top row) and acetone
(bottom row) mixing ratios obtained from a Proton-Transfer-Reaction Time-of-Flight Mass
Spectrometer (PTR-ToF-MS) aboard the NASA C130 aircraft (black, with 25[th]-75[th] percentiles in
grey). All measurements are binned at 500 m resolution and campaign-median values plotted at
the mid-point of each bin starting at 250 m above the surface. Lines show linear interpolation
between these values.

.

**Section S5: Role of sea spray**

[Figure]

**Figure S10:** Box plots of simulated ship-track campaign-median sea spray aerosol mass
concentrations for the four years of the NAAMES campaigns for simulation BASE. Red line shows
median and box limits are 25[th] and 75[th] percentiles. Outliers are shown with red plus symbol.

[Figure]

**Figure S11:** NAAMES cruise-track campaign-median marine boundary layer aerosol size
distributions from marine-influenced SEMS observations (black, with 25[th] to 75[th] percentiles in
grey) and for three GEOS-Chem-TOMAS simulations with different assumptions for the sea spray
emissions. noSS: no sea spray emissions; 3xSS: sea spray emissions scaled up by 3; BASE as
described in Table 1 and Section 2.

|  | Nov 2015 Bloom Minima | May/June 2016 Bloom Maxima | Aug/Sept 2017 Declining Phase | Mar/Apr 2018 Accumulating | Annual Mean |
|---|---|---|---|---|---|
| BASE | 0.20 | 0.33 | 0.04 | 0.28 | 0.21 |
| noSS | 0.31 | 0.24 | 0.12 | 0.28 | 0.24 |
| 3xSS | 0.05 | 0.38 | 0.14 | 0.28 | 0.21 |

**Table S4:** Mean fractional error between observations and three GEOS-Chem-TOMAS
simulations for the ship-track campaign-median aerosol size distributions shown in Supplementary
Fig. S11.

[Figure]

**Figure S12:** Vertical profiles of NAAMES campaign-median integrated SMPS observations
aboard aircraft at standard temperature and pressure (STP) for particles with diameters of 10 to
282 nm (black, with 25[th]-75[th] percentiles in grey) and at STP for three GEOS-Chem-TOMAS
simulations with different assumptions for the sea spray emissions. noSS: no sea spray emissions;
3xSS: sea spray emissions scaled up by 3; BASE as described in Table 1 and Section 2. All
measurement and model output are binned at 500 m resolution and campaign-median values
plotted at the mid-point of each bin starting at 250 m above the surface. Lines show linear
interpolation between these values.

[Figure]

**Figure S13:** Vertical profiles of NAAMES campaign-median total number concentrations for
particles with diameters larger than 3 nm (N3), 10 nm (N10) and between 3 to 10 nm (N3-N10)
from CPC observations board aircraft at standard temperature and pressure (STP) (black, with 25th-
75th percentiles in grey) and at STP for three GEOS-Chem-TOMAS simulations with different
assumptions for the sea spray emissions. noSS: no sea spray emissions; 3xSS: sea spray emissions
scaled up by 3; BASE as described in Table 1 and Section 2. All measurement and model output
are binned at 500 m resolution and campaign-median values plotted at the mid-point of each bin
starting at 250 m above the surface. Lines show linear interpolation between these values.

[Figure]

**Figure S14:** Vertical profiles of NAAMES campaign-median aerosol non-refractory sulfate and
organic mass concentrations from Aerosol Mass Spectrometer and refractory black carbon from
SP2 observations aboard aircraft at standard temperature and pressure (STP) (black, with 25[th]-75[th]
percentiles in grey) and at STP for three GEOS-Chem-TOMAS simulations with different
assumptions for the sea spray emissions. noSS: no sea spray emissions; 3xSS: sea spray emissions
scaled up by 3; BASE as described in Table 1 and Section 2. All measurement and model output
are binned at 500 m resolution and campaign-median values plotted at the mid-point of each bin
starting at 250 m above the surface. Lines show linear interpolation between these values.